# Social communication of predator-induced changes in *Drosophila* behavior and germ line physiology

Balint Z Kacsoh[1†], Julianna Bozler[1†], Mani Ramaswami[2,3], Giovanni Bosco[1]*

[1]Department of Genetics, Geisel School of Medicine at Dartmouth, Hanover, United States; [2]Smurfit Institute of Genetics, Department of Zoology, Trinity College Dublin, Dublin, Ireland; [3]Trinity College Institute for Neuroscience, Trinity College Dublin, Dublin, Ireland

**Abstract** Behavioral adaptation to environmental threats and subsequent social transmission of adaptive behavior has evolutionary implications. In *Drosophila*, exposure to parasitoid wasps leads to a sharp decline in oviposition. We show that exposure to predator elicits both an acute and learned oviposition depression, mediated through the visual system. However, long-term persistence of oviposition depression after predator removal requires neuronal signaling functions, a functional mushroom body, and neurally driven apoptosis of oocytes through effector caspases. Strikingly, wasp-exposed flies (teachers) can transmit egg-retention behavior and trigger ovarian apoptosis in naive, unexposed flies (students). Acquisition and behavioral execution of this socially learned behavior by naive flies requires all of the factors needed for primary learning. The ability to teach does not require ovarian apoptosis. This work provides new insight into genetic and physiological mechanisms that underlie an ecologically relevant form of learning and mechanisms for its social transmission.

*For correspondence: giovanni. bosco@dartmouth.edu

†These authors contributed equally to this work

## Introduction

All organisms must acquire and respond to information about their environment. Some changes in the environment are predictable or periodic, like light/dark or seasonal cycles that result in organismal adaptation manifesting as physiological changes in order to optimize survival and fitness in the context of a changing environment (*Baldwin and Meldau, 2013*; *Cermakian et al., 2013*). This ability to adapt to environmental change is essential for survival, but can such an adaptive response occur in the absence of the direct experience?

Well-defined examples of this phenomenon have been observed in what are considered 'social' organisms (*Franks et al., 2002*; *Townsend et al., 2011*). Yet, emerging studies are providing mounting evidence to suggest that the use of social cues extend far beyond the traditional notions of social animals: organisms once viewed as asocial in nature are now known to have advanced forms of social communication (*Gariepy et al., 2014*). This social transmission of information can result in distinct behavioral changes, based on another individual's set of experiences. The ability to learn from others influences the choices and behaviors of individuals and allows a group of individuals to share information about a changing environment. It is speculated that social information transmission involves either the ability to feel vicarious reward and punishment or other complex communication strategies to transmit an individual's experience to the community of conspecifics. The potential benefits of adaptive behavior, based on information acquired from others within the community, can give social learners a significant advantage over those that must directly explore and gather environmental information for themselves. Understanding how this information transfer occurs

**eLife digest** Every animal must be able to adapt to threats and changes to their environment that could affect their survival. Some 'social' animals, such as honeybees and ants, go further than this, and also transmit information about a threat—and how to survive it—to other members of their species. This helpful behavior is now known to occur to some extent even in animals that have not been considered to be social, like the *Drosophila* species of fruit fly.

Parasitoid wasps lay their eggs in the larvae and pupae of certain insect species. When the wasp eggs hatch, they feed on the host insect, eventually killing it. *Drosophila* fruit flies have evolved various behaviors to protect their offspring from these wasps. For example, female fruit flies reduce the number of eggs they lay when they are in the presence of a wasp.

Kacsoh, Bozler et al. exposed female flies to wasps for a day. These flies produced fewer eggs than flies that were not exposed to wasps and continued to lay fewer eggs for 24 hours after the wasps were removed. Introducing these flies to 'naive' flies that had not encountered a wasp caused the naive flies to produce fewer eggs as well.

After ruling out several possible ways that the wasp-exposed flies might 'teach' the naive flies to produce and lay fewer eggs, Kacsoh, Bozler et al. found that naive flies cannot learn this behavior when they are blind. In addition, exposed flies cannot instruct other flies of the threat if their wings are absent or deformed. These and other findings, therefore, suggest that information about the wasp threat is transmitted through visual cues that involve the wings.

Kacsoh, Bozler et al. found that the flies must have certain brain circuits associated with memory and learning to be able to teach others and to reduce the numbers of eggs they lay after the wasp has been removed. This suggests that signals from this brain region must be continually sent out to alter the physiology of the developing eggs in order to maintain the lower rate of egg laying; understanding how flies use visual cues for communication and how the brain signals to the ovary remain key challenges for future work.

and what the underlying neurological and molecular mechanisms are is critical for a comprehensive view of adaptive behavior across a wide range of taxa.

Many species considered as 'social' and 'non-social' communicate about the environment. Plants have been observed to alter their physiology in response to signaling from another plant (*Baldwin and Schultz, 1983*). An example of such communication involves salt stress, which has been shown to trigger the release of volatile organic compounds that induce salt resistance in neighboring plants that have yet to experience any salt stress (*Lee and Seo, 2014*). In animals, the process is speculated to be more complex: honeybees are able to fine tune signals directed at individuals within the hive that elicit highly specific behavioral changes in response to specific environmental cues (*Wenner, 1962*; *Schneider and Lewis, 2004*; *Richard et al., 2012*). Even *Drosophila* are prone to social cues, altering their decision making based on the behavior of conspecifics (*Mery et al., 2009*; *Sarin and Dukas, 2009*; *Battesti et al., 2012*). It is clear that the once thought 'fine line' between social and non-social organisms is beginning to blur, and that social communication is actually much more fundamental to life than originally considered.

In animals, this ability to transmit and process information about the environment has been termed 'social learning' (*Gariepy et al., 2014*; *Gruter and Leadbeater, 2014*). Learning can occur in a social context through olfactory cues, observation and instruction, or by imitation, and thus, is a mechanism for sharing information about a changing environment (*Baldwin and Meldau, 2013*; *Cermakian et al., 2013*). The potential benefits of adaptive behavior, based on information acquired from others within the community, can give social learners a significant advantage over those that must directly explore and gather environmental information for themselves. However, in general, the underlying molecular mechanisms of social learning are almost entirely mysterious and remain a *terra incognita* in terms of the strategies for communication, perception, neural plasticity, and the underlying physiological changes that cause changes in behavior. In this study, we use endoparasitoid wasps to explore social learning in the *Drosophila* model system with the aim of addressing some of these open questions.

Endoparasitoid wasps are ubiquitous keystone species in many ecosystems around the world. These wasps prey on immature stages of other insects, using larva and pupa of certain species as hosts

for their own offspring. Such wasps pose a serious threat to juvenile *Drosophila*, with infection rates as high as 90% in natural populations (*Janssen et al., 1988*; *Driessen et al., 1990*; *Fleury et al., 2004*). Adult *Drosophila* have evolved complex behavioral changes to protect their offspring from these predatory wasps, including altered food preference and reduced oviposition rates (*Lefevre et al., 2012*; *Kacsoh et al., 2013*). Adult *Drosophila* themselves are not infected by these wasps, thus, making the change in reproductive behavior beneficial only to an anticipated threat to their offspring and not a response to predation itself. A remarkable feature of this altered reproductive behavior is that female *Drosophila* never having seen this predator can nevertheless robustly and reproducibly respond to it, suggesting an innate recognition of this predator-threat. Here, we use this natural predator system to explore predator threat communication within *Drosophila melanogaster* and describe the specific learning, memory, and anatomical components necessary for this response. Our findings report the first example of social learning in *Drosophila* that can be delineated from simple mimicry, through the use of a natural predator. Exposure to the predatory wasp results in a distinct germ line-cell physiological apoptotic response in both flies having seen the wasp (direct experience) or flies having been paired with experienced individuals (social learning), which is clearly independent of mimicry. Furthermore, we address the genetic factors, neural circuits, and behavioral changes necessary for the transmission of this socially learned alteration in germ line physiology.

## Results

### Flies respond to wasps by decreasing oviposition and are able to confer this information to naive flies

*Drosophila melanogaster* alters its egg-laying behavior after it encounters parasitoid wasps, which infect fly larvae. This behavioral change entails at least two different and quantifiable behavioral responses. First, if high-ethanol containing food is made available to adult *Drosophila*, then wasp-exposed females will actively prefer to lay eggs on ethanol-laden food (*Kacsoh et al., 2013*). Second, if ethanol-containing food is not an option, *Drosophila* females will depress their egg-laying frequency, presumably to allow for time to search and discover an egg-laying environment that is not wasp infested (*Lefevre et al., 2012*). Adult *Drosophila* are not infected by these wasps, thus, making the change in reproductive behavior beneficial only to an anticipated threat to their offspring. To address the question of whether changes in reproductive behavior could be transferred from exposed teacher flies to naive student conspecifics, we examined the underlying physiological, physical, and genetic components of the exposed teacher and naive student flies and asked if these mechanisms rely on learned reproductive behavior.

Drosophila were exposed for 24 hr to wasps in cylindrical 7.5-cm long by 1.5-cm diameter tubes arrayed into fly condos of 24-tubes where each tube contained five female flies and one male fly, either with three female wasps (exposed) or with no wasps at all (unexposed) (*Figure 1A*, see methods and supporting information for details). After 24 hr, food plates were removed and embryos counted. Consistent with previous observations (*Lefevre et al., 2012*), exposed females reduced their oviposition rate significantly (average unexposed lay ~65 ± 3.2 eggs; average exposed lay ~13 ± 1.98 eggs) (*Figure 1B*). We observed this robust response in at least four different genetic backgrounds including *Canton-S* (CS), *Oregon-R* (OreR) (unexposed ~57 ± 2.84 eggs compared to exposed 13 ± 1.88 eggs on average), $w^{1118}$ (unexposed ~25 ± 1.54 eggs compared to exposed ~1 ± 0.53 egg on average), and transgenic flies carrying Histone H2AvD-GFP (His-GFP) (unexposed ~108 ± 7.69 eggs compared to exposed 18 ± 1.97 eggs) (*Clarkson and Saint, 1999*). To test whether this decrease in egg laying can be transmitted from exposed flies to naive females, we exposed *Canton-S* flies to wasps for 24 hr, then removed the wasps and placed these pre-exposed flies in a new condo with three naive female flies expressing Histone-GFP (His-GFP) for an additional 24 hr (*Figure 1A*). The His-GFP line was ideal for discriminating mixed populations of non-green fluorescent protein (GFP) and GFP embryos since this histone is clearly visible by 70 min after oviposition (embryonic cell cycle 9) (*Foe et al., 1993*; *Clarkson and Saint, 1999*) (*Figure 1—figure supplement 1A,B*). Oviposition in exposed teacher females was significantly reduced during the 24-hr exposure to wasps (acute depression: 0–24 hr) (53 ± 3.35 compared to 14 ± 1.59 eggs) and this depression persisted for an additional 24-hr post wasp exposure (learned depression: 24–48 hr) (35 ± 2.44 compared to 19 ± 1.33 eggs), relative to age-matched, unexposed sibling controls (*Figure 1C*, *Figure 1—figure supplement 1C*). Quantification of total GFP and non-GFP embryos deposited during the 24–48 hr after initial teacher

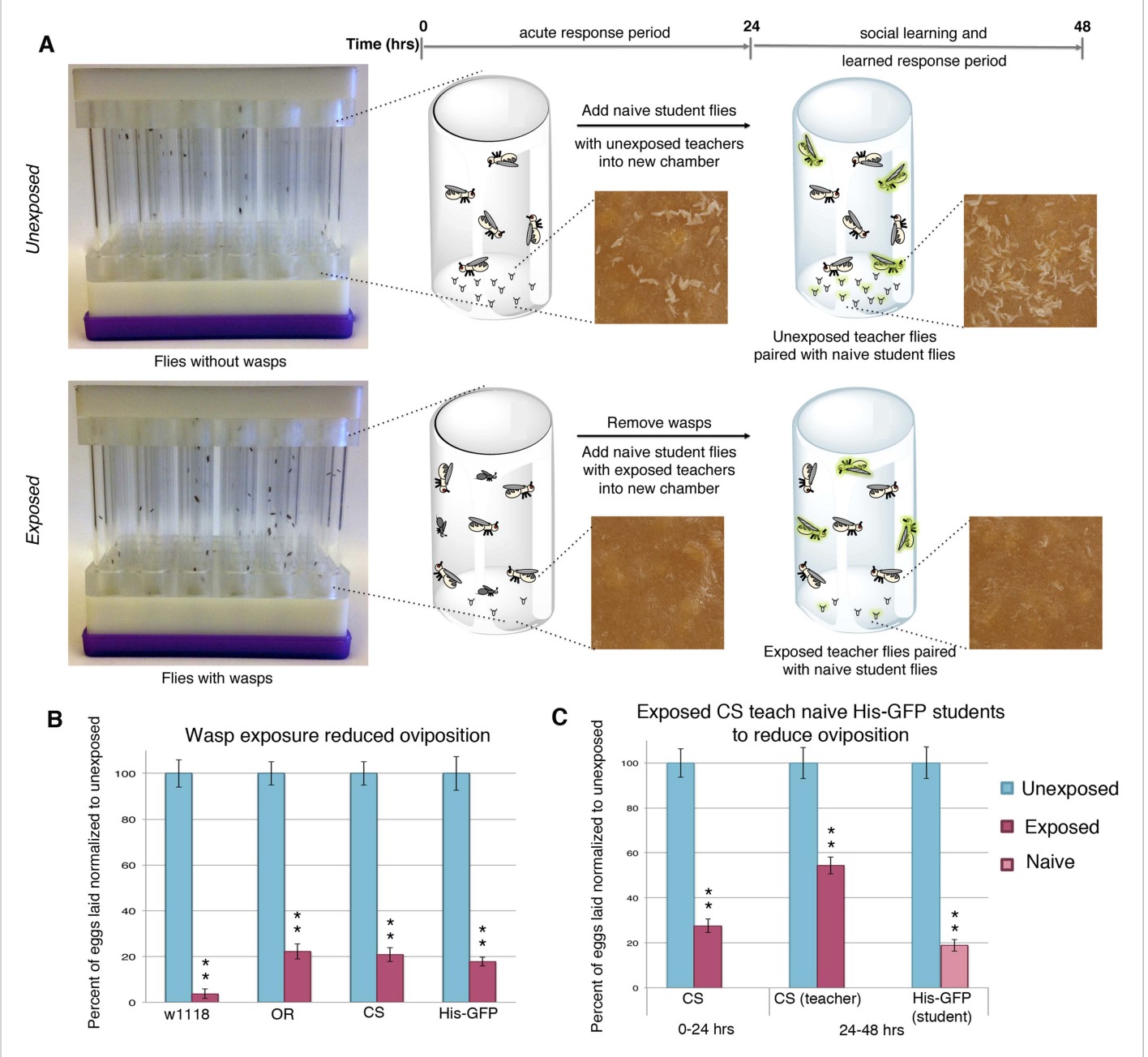

**Figure 1**. Flies respond to wasps by decreasing oviposition and are able to confer this information to naive flies. (**A**) Standard exposure setup. (**B** and **C**) Percent of eggs laid normalized to unexposed. (**B**) Wild-type flies unexposed or exposed to wasps. (**C**) *Canton-S* teachers and His-GFP students. For (**B**) and (**C**), error bars represent standard error ($n$ = 24 biological replicates) (**p < 1.0e-5).

The following figure supplement is available for figure 1:

**Figure supplement 1**. Social transmission of information from wasp-exposed female teacher fly to naive female student fly occurs.

exposure to wasps demonstrated that naive His-GFP student flies had also decreased oviposition, relative to His-GFP siblings mixed with unexposed *Canton-S* flies (33 ± 2.34 compared to 6 ± 0.86 eggs) (*Figure 1C*, *Figure 1—figure supplement 1C*). In the reciprocal experiment, naive *Canton-S* student flies mixed with pre-exposed His-GFP teacher flies also exhibited a decrease in oviposition (46 ± 2.48 compared to 14 ± 1.34 eggs, see *Supplementary files 6,7* for all raw egg numbers)

(*Figure 1—figure supplement 1D,E*). Thus, naive female flies, never experiencing wasp exposure directly, reduced oviposition when encountering exposed flies. The decrease in oviposition of student flies is not due to an effect of the ratio of teacher to student flies. We tested a 1:1 ratio of 3 exposed female teachers to 3 naive female student flies. This elicited a similar reduction in oviposition (*Figure 1—figure supplement 1F,G*). Interestingly, when we tested a 1:1 ratio of 3 exposed males to 3 naive female student flies, we found no significant decrease in oviposition for students instructed by exposed males (*Figure 1—figure supplement 1H,I*). This suggests that, under these conditions, only females can transmit predator-response information. Males are neither necessary nor sufficient for the information transfer. Therefore, for all further experiments, we used a teaching cohort of 5 females and 1 male to 3 female students, unless otherwise noted.

## Teacher-instructed student flies are unable to become teachers

To test whether the decrease in oviposition can be transmitted from students to a new batch of naive flies, we removed *Canton-S* pre-exposed teacher females from student His-GFP expressing flies and placed the teacher-instructed student flies in a new chamber with 3 new, naive *Canton-S* flies (*Figure 2A*). In teacher-instructed student flies, reduced oviposition behavior persisted for 24 hr after they were separated from teacher flies, indicative of a persisting memory of social learning. Interestingly, we found that our teacher-instructed student His-GFP flies were not able to instruct new students, as the naive *Canton-S* females did not decrease oviposition (*Figure 2B*, *Figure 2—figure supplement 1A*).

We postulated that perhaps information transfer could only occur once between wasp-exposed teachers and student flies, leading to the inability of students to further pass on information and become teachers. To test this, we removed the first cohort of student His-GFP expressing flies and placed the *Canton-S* pre-exposed teacher female flies in a new chamber with a second cohort of 3 new, naive *Canton-S* flies (*Figure 3A*). We found that oviposition depression in exposed teacher females was persistent for an additional 24-hr post wasp exposure (learned depression: 48–72 hr), relative to age-matched, unexposed, sibling controls (*Figure 3B*). Quantification of total GFP and non-GFP embryos deposited during the 48–72 hr after initial teacher exposure to wasps demonstrated that the second cohort of naive His-GFP student flies had also decreased oviposition, relative to His-GFP siblings mixed with unexposed *Canton-S* flies (*Figure 3B*). In the reciprocal experiment, a second cohort of naive *Canton-S* student flies mixed with pre-exposed His-GFP teacher flies also exhibited a decrease in oviposition (*Figure 3C*). Our results demonstrate that teachers can instruct multiple cohorts of students, thus, the inability of a student to become a teacher is not due to a limitation in amount a teacher can teach.

## Wasp exposure induces stage-specific apoptosis in wasp-exposed teachers

In order to better understand the physiological basis of how a predator-threat leads to changes in oviposition behavior, we examined the status of egg production in exposed female ovaries. Given that poor nutrition or other stressors can cause egg chambers in the ovaries to be eliminated by apoptosis at oogenesis checkpoints in region-2/3 of the germarium or stage 7/8 egg chambers (the mid-oogenesis checkpoint) (*Drummond-Barbosa and Spradling, 2001*; *McCall, 2004*), we hypothesized that the presence of parasitoid wasps could similarly reduce oviposition by triggering an oogenesis checkpoint, and thus, account for depressed oviposition. Therefore, we quantified stage-specific apoptosis in ovaries of exposed females.

Dissection of ovaries from females having been exposed to wasps for 24 hr revealed a significant increase in the number of egg chambers exhibiting apoptosis relative to unexposed sibling control females (*Supplementary file 1A,B*). Interestingly, the majority of apoptosis was observed at the stage 7/8 egg chamber checkpoint, with almost no apoptosis in region 2/3, as visualized by DNA staining with 4′, 6-diamidino-2-phenylindole (DAPI), suggesting that the pathway through which apoptosis was being triggered is fundamentally different from previously described apoptotic events (*Drummond-Barbosa and Spradling, 2001*; *McCall, 2004*) (*Supplementary file 1A,B*, *Figure 4A–F*). *Canton-S* and His-GFP fly ovaries were easily distinguishable when stained together, thus, making it possible to score apoptosis levels in ovaries of exposed and unexposed females under completely identical conditions (*Figure 4—figure supplement 1A–D*). Further confirmation that wasp exposure triggered a true apoptotic event is evidenced by the presence of characteristic

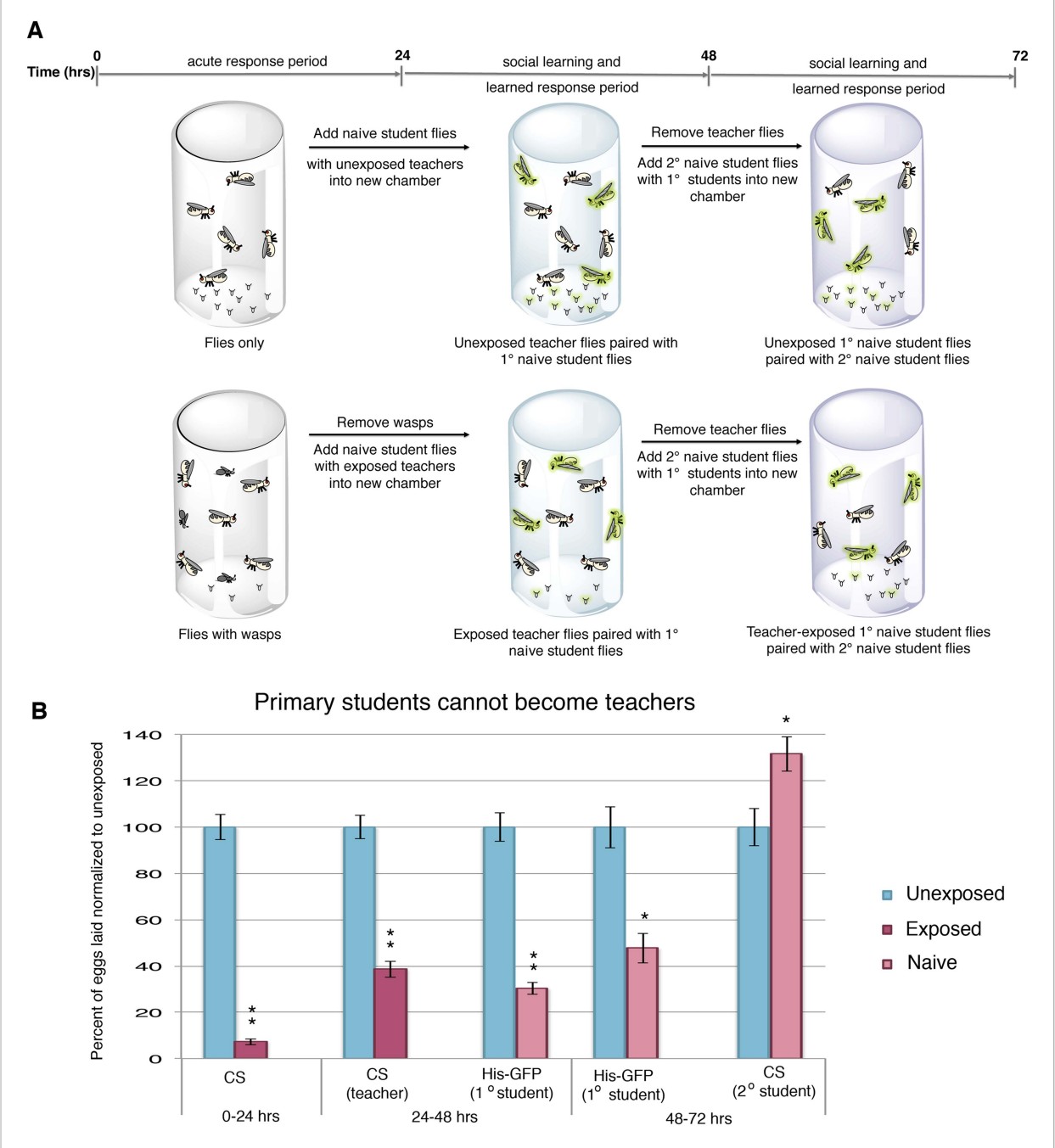

**Figure 2**. Student flies cannot become teachers. (**A**) Standard exposure setup. (**B**) Teacher exposed primary student His-GFP flies paired with naive secondary student Canton-S flies. Error bars represent standard error ($n$ = 24 biological replicates) (*p < 0.05, **p < 1.0e-5).

The following figure supplement is available for figure 2:

**Figure supplement 1**. Student flies cannot become teachers.

DAPI-intense pychnotic nurse cell nuclei, by terminal deoxynucleotide transferase dUTP nick end labeling (TUNEL) stain that detects fragmented DNA (*Figure 4G–J* and *Figure 4—figure supplement 1E,F*), and activated caspase-3 staining (*Figure 4—figure supplement 1G–J*): All positive markers of the cell death process (*McCall, 2004*). We noted that both DAPI and TUNEL were readily detected in apoptotic stage 12/13 nurse cells in both exposed and unexposed females

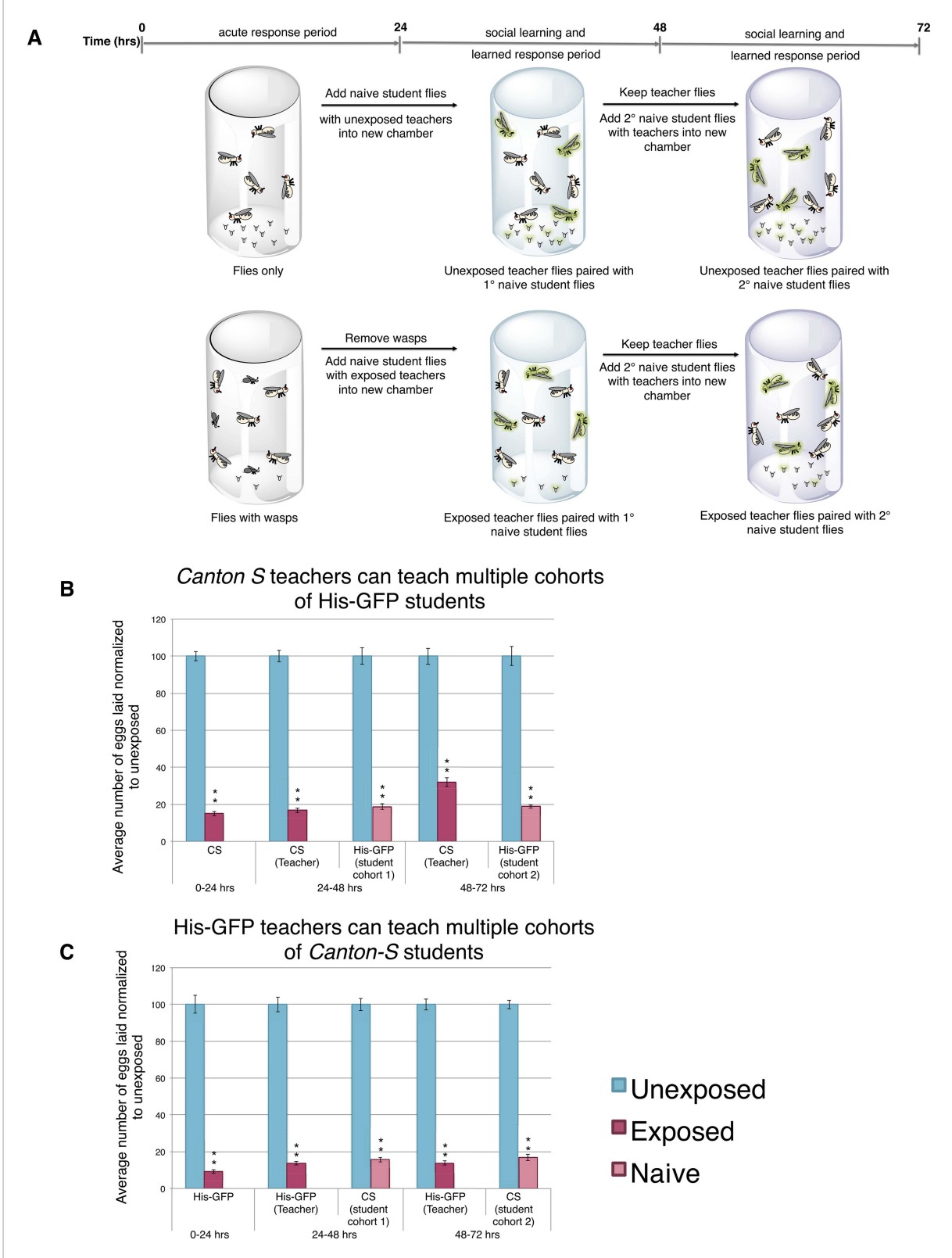

Figure 3. Teacher flies can teach multiple batches of students. (A) Standard exposure setup for teachers teaching multiple batches of students. (B and C) Percent of eggs laid normalized to unexposed. (B) Canton-S flies unexposed or exposed to wasps and paired with primary and secondary His-GFP students. (C) His-GFP flies unexposed or exposed to wasps and paired with primary and secondary Canton-S students. For (B) and (C), error bars represent standard error ($n = 24$ biological replicates) (**$p < 1.0e-5$).

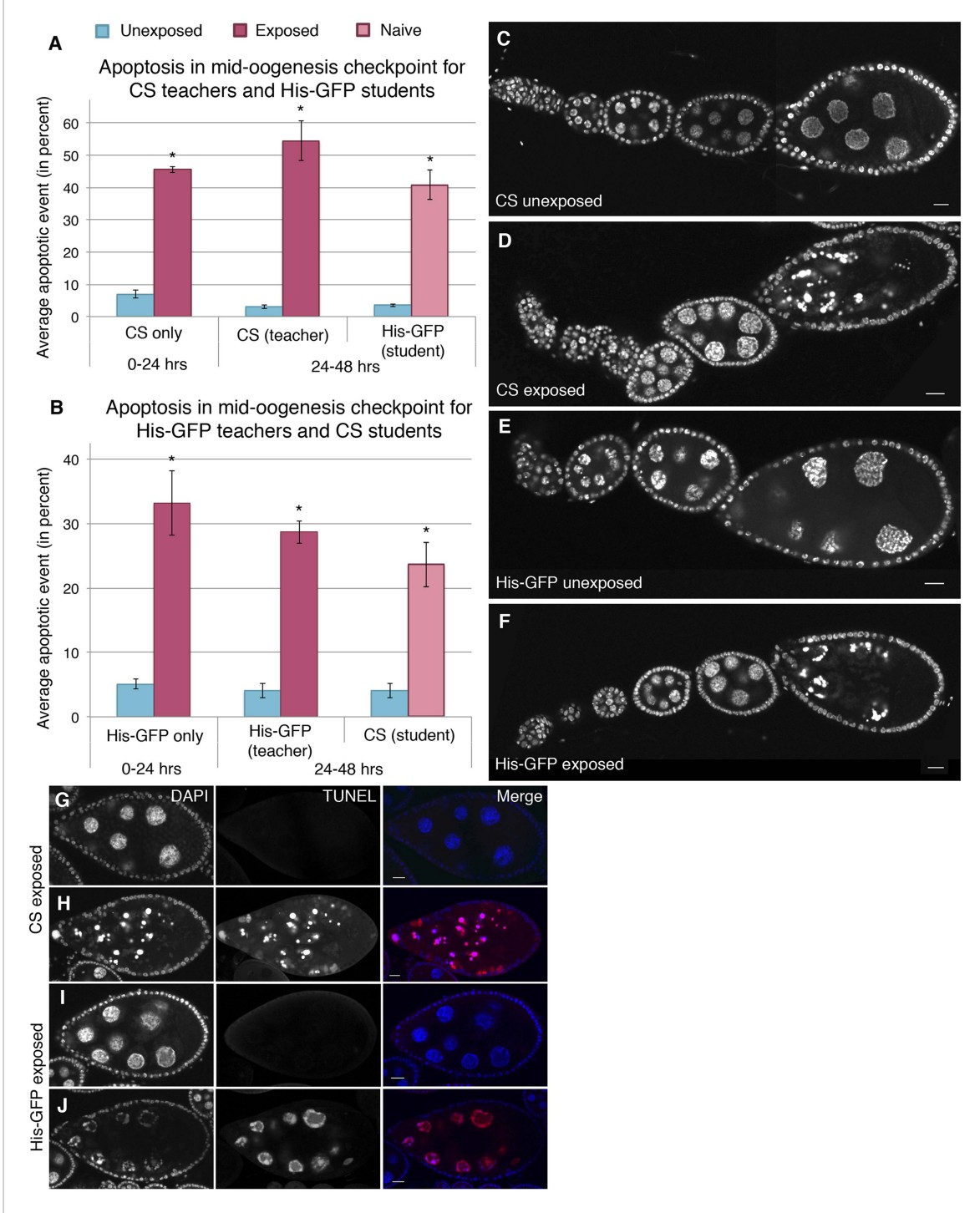

**Figure 4**. Stage-specific apoptosis observed in wasp-exposed teachers and teacher-exposed student flies. (**A** and **B**) Average percent of apoptotic events for stage 7/8 egg chambers. (**A**) Canton-S exposed and unexposed ovary apoptosis. (**B**) His-GFP exposed and unexposed ovary apoptosis. (**C** to **D**) Canton-S unexposed/exposed ovariole. (**E** to **F**) His-GFP unexposed/exposed ovariole. (**G** to **H**) Canton-S transferase dUTP nick end labeling (TUNEL) staining performed on exposed fly ovaries. (**I** to **J**) His-GFPTUNEL staining. For (**A**) and (**B**), error bars represent standard error (n = 3 biological replicates from which 12 ovaries were scored for each group) (*p < 0.05). Scale bars, 20 μm.

The following figure supplement is available for figure 4:

**Figure supplement 1**. Stage-specific apoptosis is induced following wasp exposure.

at similar levels. Developmentally regulated cell death is normally expected to eliminate late-stage nurse cells in maturing oocytes, thus, serving as an internal control for the level of detected apoptosis in exposed and unexposed females (*Supplementary file 1A,B*). Similar to the reduced oviposition behavior observed, this physiologically triggered apoptosis specifically of stage 7/8 egg chambers persisted well beyond the period of initial wasp exposure (*Figure 4A,B*, *Supplementary file 1C,D*).

## Flies continue to eat high-protein diet following wasp exposure but still depress oviposition

We considered the possibility that exposure to wasps could change fly feeding behavior, and subsequent poor nutrition could trigger the mid-oogenesis checkpoint (*Drummond-Barbosa and Spradling, 2001*). We gave both exposed and unexposed flies a high-protein yeast food stained with red food dye to visualize food intake. We found that both wasp-exposed and unexposed flies exhibited a similar amount of high-protein yeast food intake even when given a choice to feed on normal food without yeast by visualizing the red dye in the fly abdomens (*Figure 5A–D*, *Figure 5—figure supplement 1A–D*). The red yeast paste was placed on instant *Drosophila* media, which turns blue upon contact with water, allowing us to visualize whether flies are preferring high (red)- or low (blue)- nutrient food (*Figure 5—figure supplement 1E–L*). We found that even in the presence of high-protein yeast food, exposed flies still depressed oviposition when compared to unexposed controls, in addition to having apoptosis induced at the egg chamber stage 7/8 checkpoint (*Figure 5E–G*, *Figure 5—figure supplement 1M–T*, *Supplementary file 1E*). Thus, the mid-oogenesis apoptosis checkpoint triggered in exposed flies is not due to a poor nutrition intake. These data are indicative of a predator-induced neuroendocrine signaling pathway that impinges on a pathway specifically controlling mid-oogenesis specifically (stage 7/8 but not stage 2/3), and therefore, is likely different from the previously described poor nutrition oogenesis checkpoint.

## Naive student flies induce apoptosis when paired with wasp-exposed teachers

To test whether triggering of the mid-oogenesis check point could be transmitted from experienced, wasp-exposed females to naive females, we mixed teacher and student flies as described above. Naive student flies mixed with exposed teachers showed apoptosis at the stage 7/8 checkpoint, as did their teachers (*Supplementary file 1C,D,F,G*, *Figure 4A–B*). Students mixed with unexposed, 'mock' teachers did not show significant levels of increased apoptosis in the ovary (*Supplementary file 1C,D,F,G*, *Figure 4A–B*). Thus, in naive student flies, transmitted information from exposed teacher flies results in triggering a specific-apoptotic mid-oogenesis checkpoint in students that have learned from teachers' experience. These data indicate that teacher flies transmit instructive cues to student flies that student flies receive these cues and then process them in a manner that leads to apoptosis of egg precursor cells and reduced oviposition.

## Oviposition depression in teacher and student requires the caspase encoding genes Dcp-1 and drice, which are dispensable for teacher behavior

One explanation for social learning could be that student flies instinctively mimic the behavior of more experienced teacher flies. Repeated episodes of imitative behavior could lead to a strengthening of neural circuits that underlie this behavior. We explored this idea by testing if wasp-exposed flies that are genetically unable to suppress oviposition efficiency are still able to successfully act as teacher flies. The *Drosophila* mid-oogenesis checkpoint is known to activate effector caspases Dcp-1 and drice (*McCall, 2004*). Additionally, the caspase-3 staining we performed on wasp-exposed teacher ovaries recognizes effector caspases Dcp-1 and drice (*Figure 4—figure supplement 1G–J*), leading us to hypothesize that these caspases are important in oviposition depression in teacher and student flies as a response to parasitoid wasps. By using a maternal $\alpha$-Tubulin > Gal4 driver to express an RNA-hairpin targeting mRNA from each of these genes, we were able to reverse both the decrease in oviposition as well as the increase of stage 7/8 egg chamber apoptosis of wasp-exposed females, while RNAi depletion of these caspases had no effect on oviposition of unexposed females (*Figure 6—figure supplement 2A,B*). This provides further evidence that the stage 7/8 egg chamber apoptosis and corresponding oviposition decrease is a specific physiological checkpoint, similar to that previously

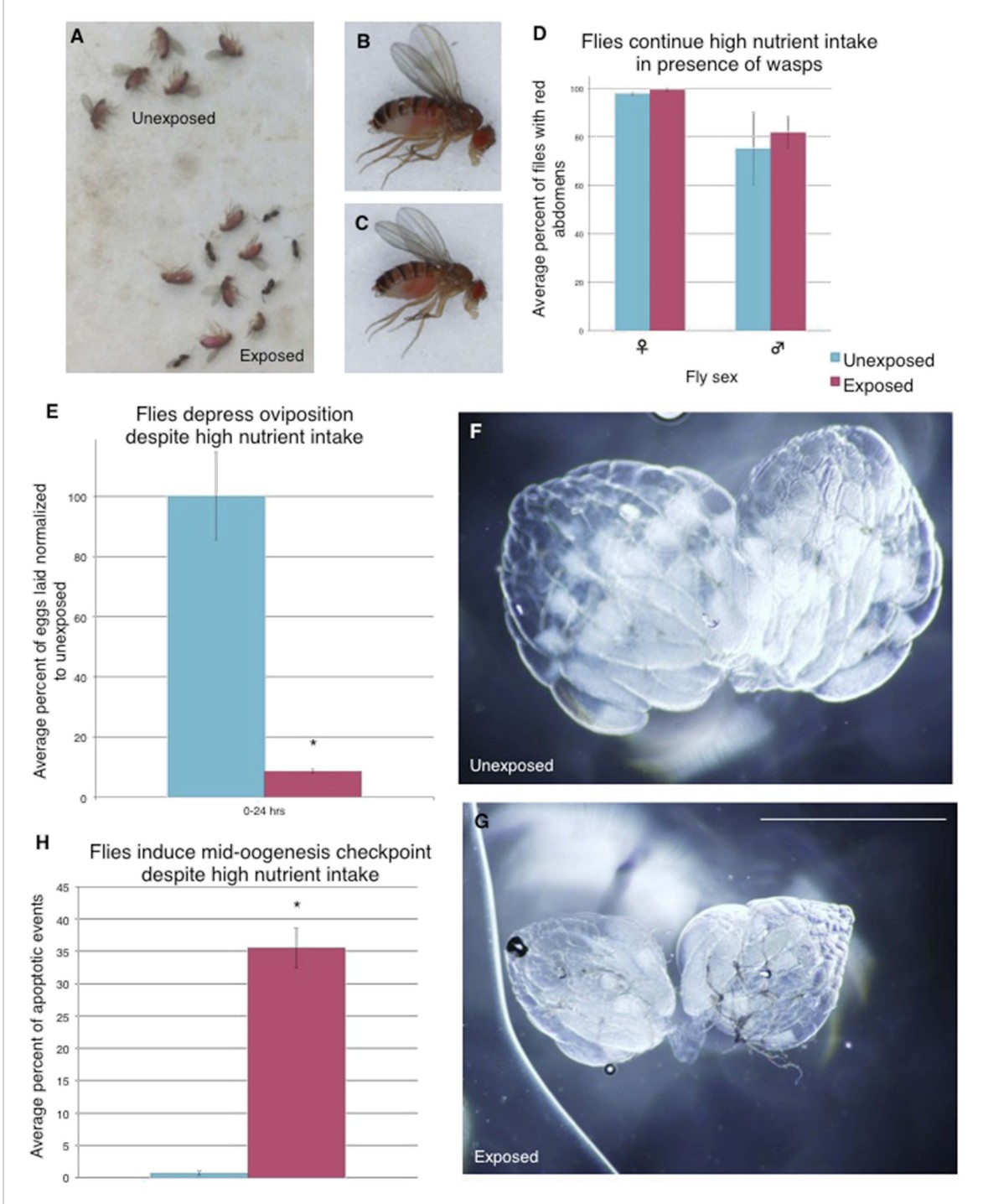

**Figure 5**. Flies continue to eat high-protein diet following wasp exposure but still depress oviposition. Continued oviposition depression cannot be explained by a lack of nutrient intake that normally inactivates insulin signaling. The high-nutrient intake by exposed female flies suggests that an active insulin signaling pathway is inhibited or bypassed downstream of nutrient sensing. (**A**) Exposed and unexposed flies anesthetized immediately after 24-hr exposure period shows red food in abdomens. (**B**) Lateral view of unexposed fly. (**C**) Lateral view of exposed fly. (**D**) Percent of male and female flies with red food in abdomen, error bars are 95% confidence intervals. (**E**) Percent of eggs laid normalized to unexposed following 24-hr exposure period. All eggs on the food plate were counted, including eggs on the yeast paste. (**F**) Representative ovary dissected from unexposed fly. 36 total ovaries were dissected and examined across 3 replicates for each treatment. (**G**) Ovary dissected from exposed fly. Scale bar for (**F**) to (**G**) is 1.0 mm. (**H**) Average percent apoptosis in mid-oogenesis checkpoint for

*Figure 5. continued on next page*

*Figure 5. Continued*

unexposed and exposed Canton S. For (**D**), (**E**), and (**H**), error bars represent standard error (n = 3 biological replicates. For (**D**), 100 female and 20 male flies were counted per replicate. For (**E**), 3 egg lay plates were counted per treatment. For (**H**), 3 biological replicates from which 12 ovaries were scored for each group) (*$p < 0.05$).
The following figure supplement is available for figure 5:

**Figure supplement 1**. Flies continue to eat high protein diet following wasp exposure but still depress oviposition.

described for poor nutritional intake (*Figure 4A,B*, *Supplementary file 1H*) (*Drummond-Barbosa and Spradling, 2001*). We considered the possibility that ovarian apoptosis could produce secondary signals important for conveying information to naive flies, which in turn triggers apoptosis in student ovaries. To test this, we used teacher flies that were incapable of triggering apoptosis because of RNAi depletion of Dcp-1 or drice, specifically in developing egg chambers. Strikingly, following wasp exposure, flies, depleted of germ line Dcp-1 or drice function, were still excellent teachers capable of cueing naive student flies to decrease their oviposition and induce apoptosis at the stage 7/8 mid-oogenesis checkpoint in the students' ovaries (*Figure 6A,B*, *Figure 6—figure supplement 1 A-B and G*, *Supplementary file 1I,J*). The finding that Dcp-1 and drice deficient females incapable of depressing oviposition can nevertheless convey critical cues to naive students demonstrates that the depressed oviposition response can be decoupled from the process required for teacher–student information transfer. Thus, information transfer in this context is not due to secondary effects of ovarian cell death. Interestingly, Dcp-1- and drice-deficient student females could not depress oviposition in response to exposed, wild-type teachers, suggesting that the same effector caspases activated in exposed teachers are also needed for oviposition depression in students (*Figure 6C–D*). Control, parental lines were found to behave as wild type as both teachers and students (*Figure 6—figure supplement 1A–F*). We tested two additional Dcp-1 (*Dcp-1²* and *Dcp-1³*) (*Etchegaray et al., 2012*) mutant lines that displayed the same phenotype as the RNAi result (*Figure 6E–F*, *Figure 6—figure supplement 1G,H*). We conclude that the depressed oviposition in student flies cannot be from simple mimicry.

## Teacher flies communicate information to naive flies through visual cues

Previous work has demonstrated that wasp-exposed females actively prefer to lay eggs on ethanol-laden food through the use of visual cues. These visual cues were important for wasp perception and subsequent behavior change (*Kacsoh et al., 2013*). Therefore, to better understand the mechanism through which information was being transferred from teacher to student flies, we tested the role of both smell and vision in information acquisition in our system by testing these mutations in both teacher and student flies. The gene *Orco* is known to be expressed in almost all olfactory receptor neurons, and the mutant-lacking *Orco* is unable to respond to smell stimuli (*Vosshall et al., 1999*). We found that *Orco¹* flies could respond to wasps and teach student flies (*Figure 7A*). Additionally, *Orco¹* flies as naive students could learn normally from teacher flies (*Figure 7B*). These data suggest that olfaction is not necessary to perceive the wasp threat nor to confer or receive the information during social learning.

We then analyzed the role of vision in this paradigm with the use of flies mutant for *ninaB*. *ninaB* is part of a single enzyme family, which acts as a key component for visual pigment production and vision in *Drosophila* (*von Lintig et al., 2001*; *Voolstra et al., 2010*). The *ninaB^P315* blind females exhibited no initial response to the presence of wasps and were not able to transmit information to naive flies (*Figure 7C*). In contrast to *Orco¹* flies, blind *ninaB^P315* student flies were unable to learn from teacher flies (*Figure 7D*).

Our *ninaB^P315* data suggest that visual stimuli are responsible for both the acute and learned response. Therefore, we wanted to further elucidate the role of vision in this system. As in previous studies, we impaired vision of wild-type flies simply by running trials in complete darkness (*Tompkins et al., 1982*; *Budick et al., 2007*; *Duistermars et al., 2009*; *Robie et al., 2010*; *Ofstad et al., 2011*). We found that performing the entirety of experiment in darkness using *Canton-S* or His-GFP teachers yielded no response to the presence of wasps and exposed females were not able to transmit information to naive flies (*Figure 8A,B*, *Figure 8—figure supplement 1A*). Similarly, performing only

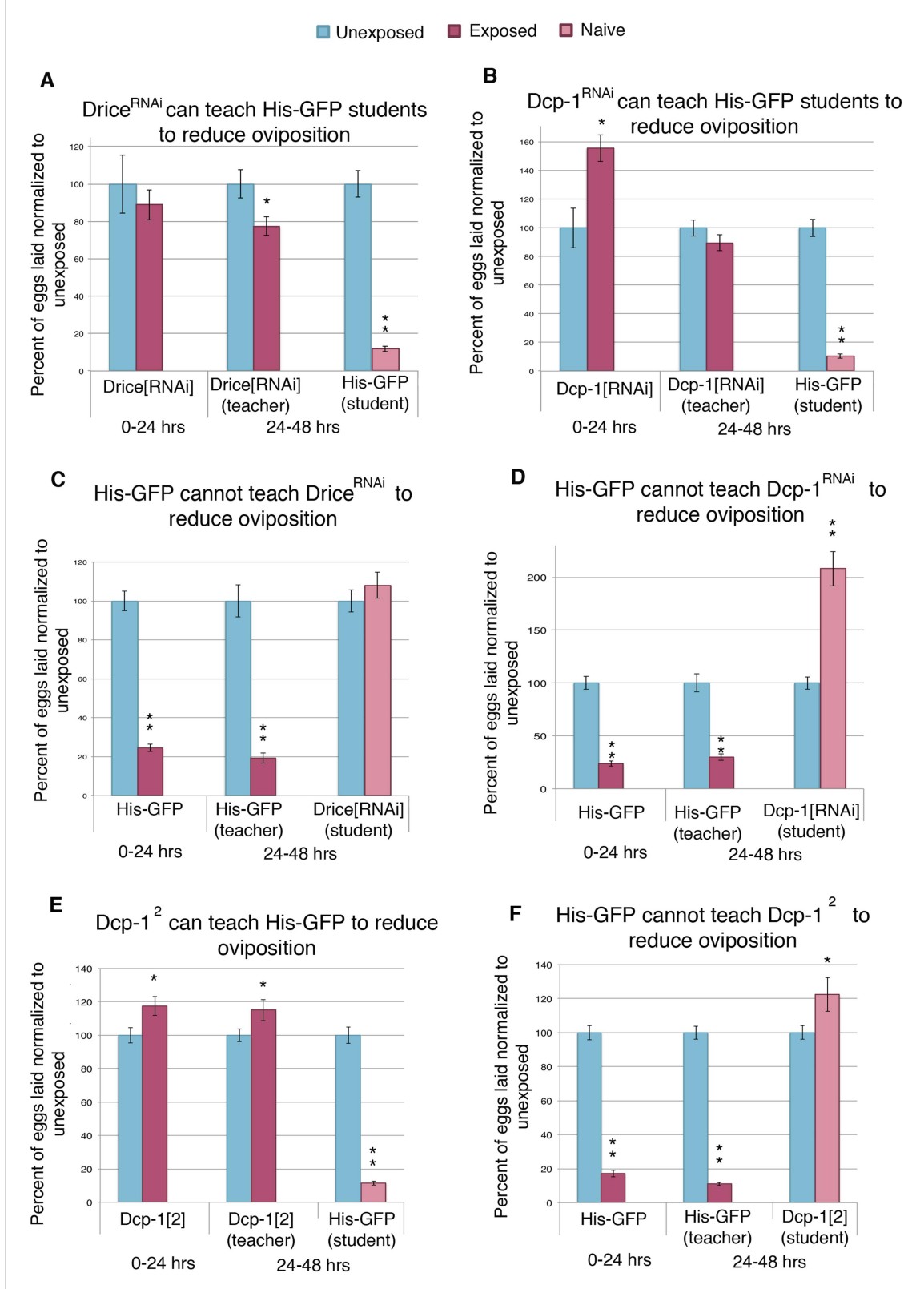

**Figure 6**. Socially transmitted oviposition depression in response to wasp exposure acts through the mid-oogenesis checkpoint. (**A** to **F**) Percent of eggs laid normalized to unexposed. (**A** and **C**) Drice RNAi-knockdown as teachers and students. (**B** and **D**) Dcp-1 RNAi-knockdown as teachers
*Figure 6. continued on next page*

*Figure 6. Continued*

and students. (**E** to **F**) Dcp-1$^2$ as teachers and students. For (**A**) to (**F**), error bars represent standard error (n = 24 biological replicates) (*p < 0.05, **p < 1.0e-5).

The following figure supplements are available for figure 6:

**Figure supplement 1**. Socially transmitted oviposition depression acts through the mid-oogenesis checkpoint.

**Figure supplement 2**. Further evidence indicating that oviposition depression acts through the mid-oogenesis checkpoint.

the wasp exposure period in the dark and the social-learning period in the light again yielded no response to the presence of wasps and these exposed females were not able to transmit information to naive flies (*Figure 8C,D*, *Figure 8—figure supplement 1B*). Finally, we performed the wasp exposure period in the light, but moved the teachers paired with students for the social-learning period into the dark (*Figure 8E*). Here, we find teacher flies had both an acute and learned response, but these teachers were not able to transmit information to naive flies, presumably due to the learning period being in the dark (*Figure 8F*, *Figure 8—figure supplement 1C*). Consistent with previous studies indicating the necessity of light in visual learning (*Ofstad et al., 2011*), these data suggest that wild-type fly vision can only detect cues from wasps and teachers if there is light present, again demonstrating the role for visual cues for the behavior.

Finally, we wanted to elucidate if a visual cue alone is sufficient to elicit the behavioral changes. Previous experiments had both teachers and students co-habitating, leading us to speculate whether other stimuli were involved in either the acute- or social-learning response. To test this, we built the Fly Duplex, which we constructed by using three standard 25 mm × 75-mm glass microscope slides that were adhered between two 75 mm × 50 mm × 1-mm glass microscope slides using clear aquarium silicone sealant, making two compartments separated by one 1-mm thick glass slide. This setup allows flies to see other flies or wasps in the neighboring chamber, but do not allow direct contact (*Figure 9A*). We find that both the acute and learned response are intact when performing the exposure in separate, but adjacent, chambers using the Fly Duplex (*Figure 9B–C*). We also find that teachers are able to transmit information to naive flies when in separate chambers, yielding depressed oviposition (*Figure 9B–C*). Both the requirement for light and the use of the Fly Duplex strongly suggest that olfactory, auditory, and tactile information is not likely to be important for this type of social communication. Instead, this demonstrates that visual cues alone are sufficient for acute-, learned-, and social-learning responses.

Collectively, our data demonstrate that teacher flies respond to a visual stimulus during wasp exposure and subsequently provide visual cues, which student flies process in a manner that leads to reduced oviposition.

## Teacher flies communicate information to naive flies using their wings

In order to elucidate the visual cue used to transmit information from teachers to naive students, we tested flies that were missing wings, either through genetic or mechanical perturbation. We first tested flies mutant in the wingless gene (*wg$^1$*). The wingless phenotype in the *wg$^1$* stock is not fully penetrant. The progeny of *wg$^1$* parents are comprised of flies with two wings, one wing, and no wings (*Figure 10A,B*, *Figure 10—figure supplement 1A,B*). Reported segregation patterns suggest that the three phenotypes are genotypically similar and that phenotypic change is a result of incomplete penetrance (*Sharma, 1973*). We find that both one-winged and two-winged mutants have an intact acute and learned response following wasp exposure (*Figure 10C*, *Figure 10—figure supplement 10F*). However, one-winged *wg$^1$* flies are unable to act as teachers, suggesting a role for both wings in communication (*Figure 10C*). Two-winged *wg$^1$* flies behaved as wild-type teachers, demonstrating that the *wg$^1$* mutation does not induce impaired teaching (*Figure 10—figure supplement 1F*). For additional validation of this observation, we mechanically removed the wings of wild-type flies. The wings of wild-type *Canton-S* flies were cut prior to wasp exposure and tested for oviposition response. These flies displayed an intact acute and learned response, but they were unable to teach (*Figure 10D–F*, *Figure 10—figure supplement 1C,D*). Finally, we used the GAL4/UAS system to express the cell death protein reaper (UAS-Rpr) in conjunction with a wing driver (MS1096) to ablate

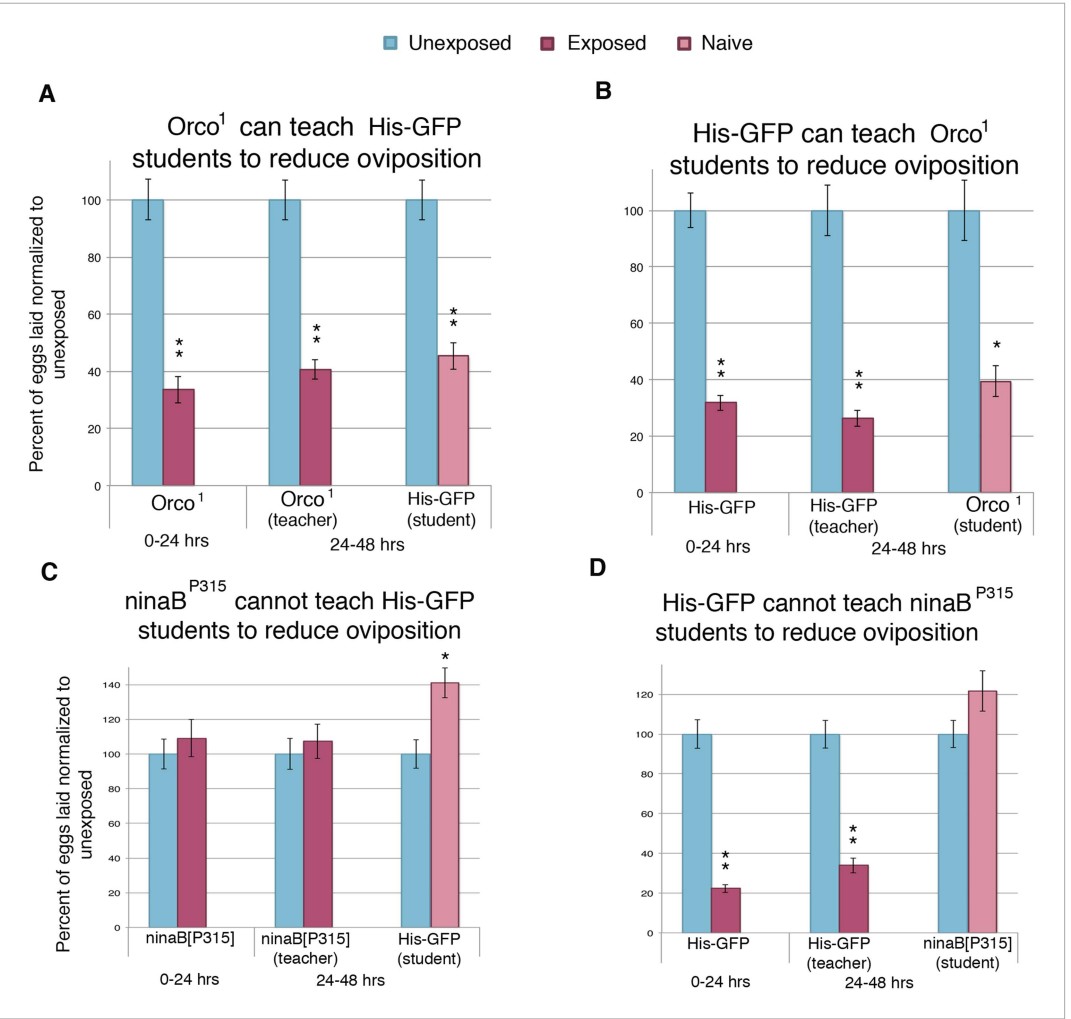

**Figure 7**. Flies respond to wasps and confer this information to naive flies through visual cues. (**A** to **D**) Percent of eggs laid normalized to unexposed. (**A** to **B**) Smell mutants as teachers and students. (**C** to **D**) Sight mutants as teachers and students. For (**A**) to (**D**), error bars represent standard error ($n = 24$ biological replicates) (*$p < 0.05$, **$p < 1.0e-5$).

proper wing development (*Figure 10G*, *Figure 10—figure supplement 1E*). We find that these flies also have an intact acute and learned response, but they were unable to teach (*Figure 10H*). Flies lacking wild-type wings were able to function as students, demonstrating that wings are not necessary for student learning (*Figure 10—figure supplement 1G*).

We hypothesized that perhaps flies whose wings had been genetically ablated or mechanically removed could be experiencing overall mobility impairment, thus, yielding the inability to teach. We decided to perform our assay using flies mutant in the *erect wing* locus, which encodes a protein, EWG. Loss-of-function *erect wing* alleles result in embryonic lethality. Viable alleles of *erect wing* cause severe abnormalities of the indirect flight muscles (*DeSimone et al., 1996*). Flies carrying viable allelic combinations of mutations at the *erect wing* (*ewg*) locus do not have, or have greatly reduced, indirect flight muscles (*Deak II et al., 1982*; *Fleming et al., 1982*). We tested two EWG alleles, *ewg[1]* and *ewg[2]*, and found that these flies displayed an intact acute and learned response, but they were unable to teach. These mutants exhibited a wild-type ability to learn from His-GFP teachers, again demonstrating that wings are not required to learn (*Figure 11A–D*). EWG is also required in the development of the nervous system (*Fleming et al., 1982*; *DeSimone and White, 1993*). Given this information, we wanted to examine if nervous system-specific expression of wild-type EWG protein in an *ewg* mutant background is sufficient to restore teaching ability. This expression does not rescue

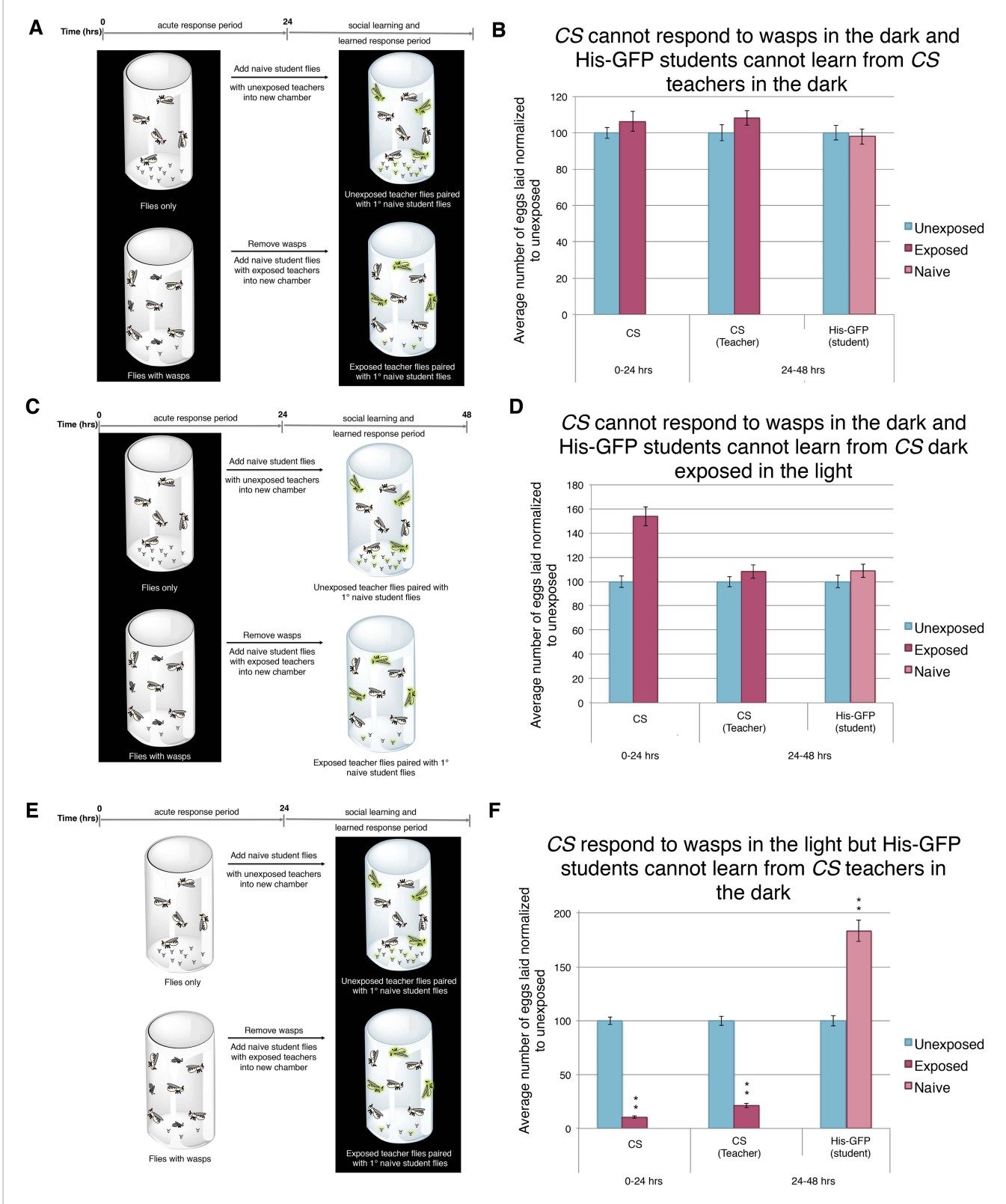

**Figure 8**. Acute and teaching response requires light. (**B**, **D**, and **F**) Percent of eggs laid normalized to unexposed. (**B**, **D**, and **F**) *Canton S* as teachers and His-GFP as students. (**A**) Exposure setup when both acute and social response occurs in dark. (**B**) Results of experiment as described in (**A**). (**C**) Exposure setup when acute response occurs in the dark but social response occurs in the light. (**D**) Results of experiment as described in (**C**). (**E**) Exposure setup

*Figure 8. continued on next page*

*Figure 8. Continued*

when acute response occurs in the light but social response occurs in the dark. (**F**) Results of experiment as described in (**E**). For (**B**), (**D**) and (**F**), error bars represent standard error (*n* = 24 biological replicates) (**p < 1.0e-5).

The following figure supplement is available for figure 8:

**Figure supplement 1**. Further evidence indicating that learning requires light.

the muscle phenotype (*DeSimone et al., 1996*). We found that *ewg^NS4* (neuronal rescue) displayed an intact acute and learned response, had the ability to learn from His-GFP teachers, but they were unable to teach (*Figure 11E–F*).

Through the use of multiple genetic mutants and genetic and mechanical perturbations of wings, we find that both wings and wing movements are necessary for teaching ability. Collectively, these data suggest that teacher flies are using their wings as the visual cue to inform naive student flies.

## Maintained oviposition and social learning require active learning and associated plasticity

To examine the possibility that the behavioral response to predator-threat requires active learning and associated plasticity in wasp-exposed flies, we asked how predator responses were affected in learning and memory mutants *rutabaga* (*rut^1*, *rut^2080*), *dunce* (*dnc^1*, *dnc^ML*), *Adf1* (*Adf1^nal*), *amnesiac* (*amn^1*, *amn^X8*), FMR1 (*Fmr1^3*, *Fmr1^B55*), and *Orb2^ΔQ*; the last being of particular significance as the *ΔQ* mutation leaves all essential functions of the Orb2 neuronal regulator intact, but deletes a Gln-rich prion domain exclusively required for persistent long-term memory, possibly by enabling an Orb2 conformational switch that leads to active synaptic translation (*Si et al., 2003*; *Keleman et al., 2007*; *Majumdar et al., 2012*). Each of these mutants responded acutely to predator presence with a dramatic decrease in oviposition when in the presence of wasps for the first 24 hr (*Figure 12A,C,E,G,I,K* and *Figure 12—figure supplement 1A,B,E,G*). This indicates that the acute oviposition depression is independent of these gene functions. However, when wasps were removed and mutant flies were placed in a new tube for an additional 24 hr after wasp exposure, oviposition returned to levels comparable to unexposed flies (*Figure 12A,C,E,G,I,K* and *Figure 12—figure supplement 1A,B,E,G*). This indicates that although the acute response to a predator threat does not require memory consolidation, the persistence of decreased oviposition behavior after wasp removal requires a form of long-term memory whose consolidation requires cAMP signaling and translational control mediated at least in part through the prion domain of *Orb2*. These results are consistent with other wasp-induced fly memory formation, specifically with respect to seeking ethanol-laden substrates upon wasp exposure (*Kacsoh et al., 2015*). Naive wild-type student flies encountering the pre-exposed mutants also did not respond through oviposition decrease (*Figure 12A,C,E,G,I,K* and *Figure 12—figure supplement 1A, B,E,G*). Collectively, the data from multiple alleles of multiple mutants indicated that these mutations yielded flies that did not retain physiological effects of the threat-response necessary to successfully transmit information to naive wild-type student females.

Unexpectedly, socially learned depression of oviposition in naive student flies was defective in *rut*, *dnc*, *Adf1*, *amn*, FMR1, and *Orb2* mutants (*Figure 12B,D,F,H,J* and *Figure 12—figure supplement 1C,D,F,H*). As these learning mutants show normal acute oviposition depression in response to direct wasp exposure, this suggests that wasp-induced and teacher-induced reductions in oviposition behavior occur through fundamentally different mechanisms. This is consistent with the fact that wasps and teachers must provide different visual signals to initiate learning and must, therefore, be expected to alter behavior through different neural circuit mechanisms. Taken together with the observations of blind *ninaB^P315* mutants, experiments performed in the dark, and the Fly Duplex, these results demonstrate that during social learning student flies must be able to visually perceive information from teacher flies and then undergo an active-learning process in order to stably respond by depressing oviposition.

We further asked how apoptosis in egg chambers was affected in wasp-exposed *orb2^ΔQ* mutant flies. The apoptotic response to acute wasp exposure (0–24 hr) in *orb2^ΔQ* was similar to the wild type, as expected, given that these flies had a normal depressed oviposition in presence of wasps (*Figure 12M*, *Supplementary file 1K*). However, in the 24-hr period following removal of wasps

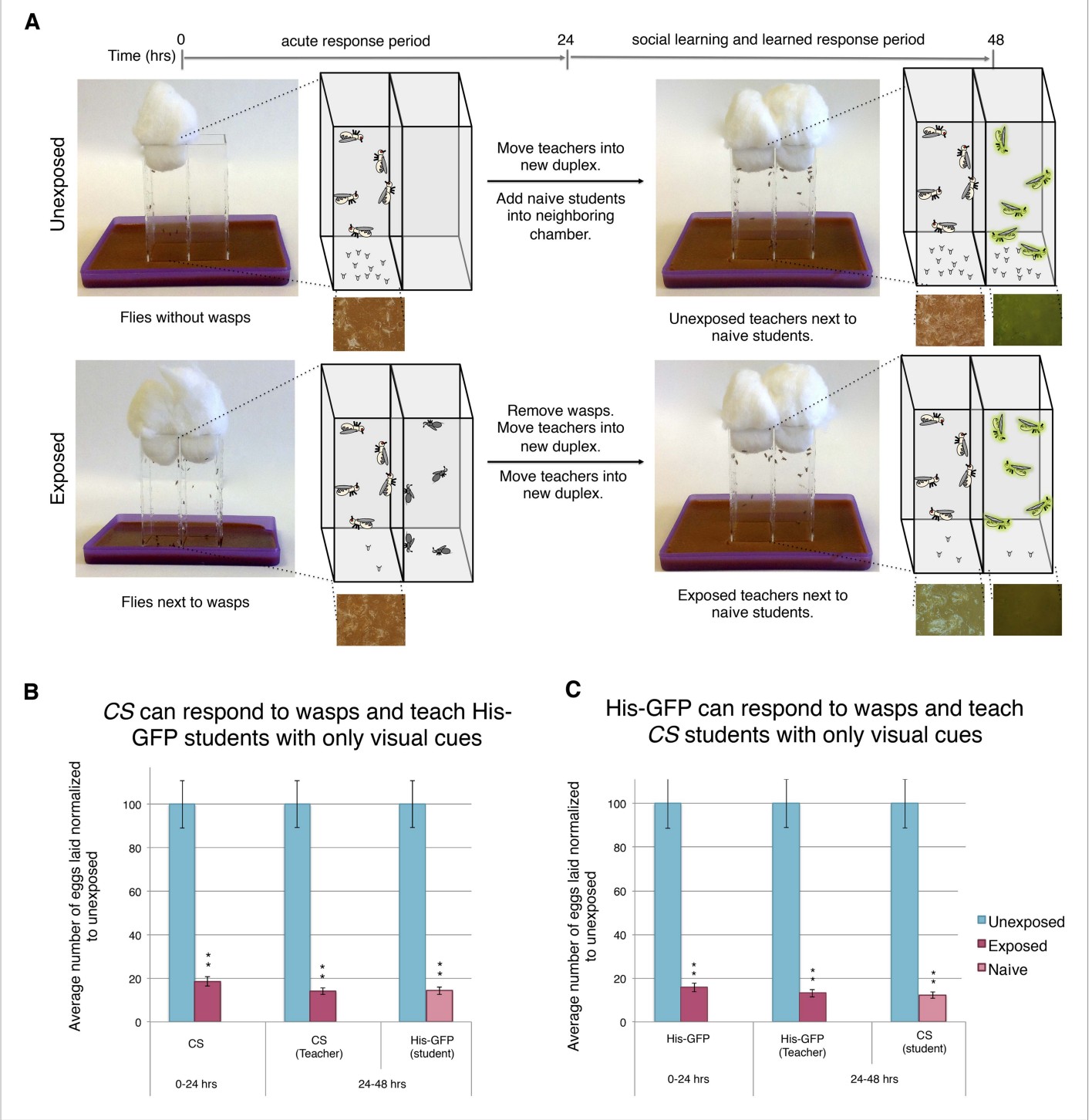

**Figure 9**. Visual cues are necessary and sufficient for learning. (**B** and **C**) Percent of eggs laid normalized to unexposed. (**A**) Standard exposure setup using the Fly Duplex. The Fly Duplex ensures only visual cues are transferred between groups. (**B**) *Canton S* as teachers with His-GFP students. (**C**) His-GFP as teachers with *Canton S* as students. For (**B** and **C**) error bars represent standard error ($n$ = 10 biological replicates) (**$p < 1.0e-5$).

(24–48 hr), orb2$^{\Delta Q}$ female flies had increased their egg laying and showed low levels of apoptosis in stage 7/8 egg chambers comparable to control unexposed flies (*Figure 12M*, *Supplementary file 1L*). We conclude that *Drosophila* females depress their egg laying during exposure to predatory wasps through an acute pathway that requires visual perception of wasp presence and leads to active

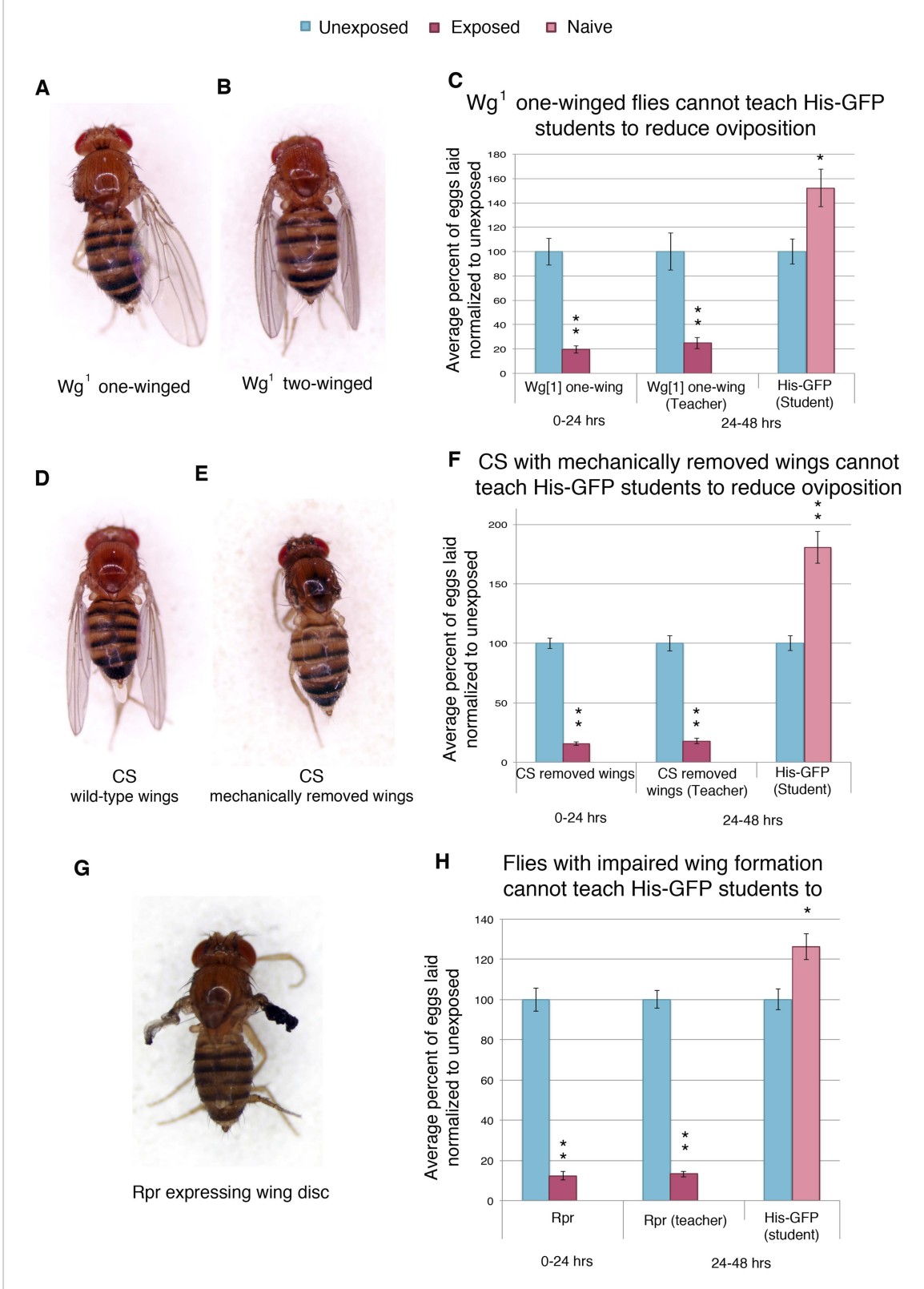

Figure 10. Teacher–student dynamics require wings to allow for communication to take place. (For **C**, **F**, and **H**) Percent of eggs laid normalized to unexposed. (**A**) Dorsal view of *wg*[1] with one wing. (**B**) Dorsal view of *wg*[1] with two wings. (**C**) *wg*[1] one-winged flies as teachers. (**D**) Dorsal view of Canton-S female. (**E**) Dorsal view of Canton-S female with clipped wings. (**F**) Canton-S flies with clipped wings as teachers. (**G**) Dorsal view of a female fly expressing
*Figure 10. continued on next page*

*Figure 10. Continued*

reaper in the wing disc. (**H**) Flies expressing reaper in the wing disc as teachers. Error bars represent standard error (For (**C**) n = 18 biological replicates.) (For [**F** and **H**] n = 24 biological replicates) (\*p < 0.05, \*\*p < 1.0e-5).

The following figure supplement is available for figure 10:

**Figure supplement 1**. Teacher flies need wings in order to instruct student flies.

elimination of developing eggs. The persistence of depressed oviposition and apoptosis in the 24-hr period after wasp removal requires an intact *orb2* gene, suggesting that maintenance of the initial behavior may require neural consolidation of the memory of wasp presence learned during the exposure period. Both acute and persistent mechanisms indicate that a systemic pathway initiated in photoreceptors and visual systems of female flies, processed centrally through neural circuits that can encode memories, leads to neuroendocrine signaling that impinges on developing egg chambers where it activates caspase-signaling cascades.

## Continued input from the mushroom body is required for the learned response and teaching behavior

To test if the reduced oviposition requires continued neuronal input to maintain reduced oviposition and teaching behavior, we mechanically removed neural input of exposed wild-type flies. Following wasp exposure, we surgically removed fly heads and paired them with naive student flies. Decapitated flies are of standard use in behavioral assays, and only decapitated flies that recovered after anesthesia were used (*Cook, 1975*; *Nilsen et al., 2004*; *Clyne and Miesenbock, 2008*; *Trott et al., 2012*). We found that decapitated flies could not maintain the same level of reduced oviposition as normal flies (i.e., decapitation led to an increase in oviposition), and they could no longer teach, suggesting a continued input from the brain is needed to elicit these behavioral changes (*Figure 13A–C*, *Figure 13—figure supplement 1E,F*). To ask whether the mushroom body (MB) specifically plays a role in maintained oviposition reduction and the teaching behavior, we used the GAL4/UAS system to express tetanus toxin light chain (*UAS-TeTx*) in conjunction with a MB driver (*OK-107-GAL4*) (*Aso et al., 2009*) to block synaptic transmission (*Martin et al., 2002*). The tetanus toxin light chain works by catalytically inhibiting synaptic transmission once present in the cytosol by cleaving either synaptobrevin, syntaxin, or SNAP-25 (*Poulain et al., 1988*; *Bittner et al., 1989*; *Mochida et al., 1990*; *Kurazono et al., 1992*; *McMahon et al., 1993*). We found that flies expressing *UAS-TeTx* in the MB exhibited a wild-type acute response, suggesting that the acute response occurs independent of the MB. However, in the learned period, these flies no longer showed reduced oviposition and were unable to teach naive students (*Figure 13D*). Using a second MB driver (MB247), this result was recapitulated (*Figure 13—figure supplement 1I*) (*Mao et al., 2004*). Control parental lines functioned as both wild-type students and teachers both as homozygotes and when outcrossed to *Canton-S* (*Figure 13—figure supplement 1A–H*). Flies expressing UAS-TeTx in the MB failed to function as students (*Figure 13—figure supplement 1J–K*). These data suggest that wasp presence is sensed through the visual system, and this information is relayed to the MB to induce a persistent reduction of oviposition, apoptosis, and teaching behavior, all of which are maintained over a time span of days.

## Inhibition of a canonical long-term memory gene in the MB eliminates teaching ability

We found that mutants in *orb2* exhibited a defect of oviposition depression as well as teaching and social-learning ability (*Figure 12A–B*). However, these experiments could not exclude the possibility that *orb2* gene product was required in non-neural tissues. Similarly, *orb2* may have been necessary for early neuronal development, and mutant phenotypes observed simply reflected developmental defects that precluded proper adult MB functions (pleiotropic effects). Given that inhibiting synaptic transmission in the MB with *UAS-TeTx* eliminated a long-term behavioral response to wasp exposure, teaching ability, and social learning (*Figure 13D*, *Figure 13—figure supplement 1I–K*), we tested the hypothesis that the gene products of known learning and memory genes (such as *orb2*) may also be required to function in this anatomical region of the brain. To test this, we used the GAL4/UAS system

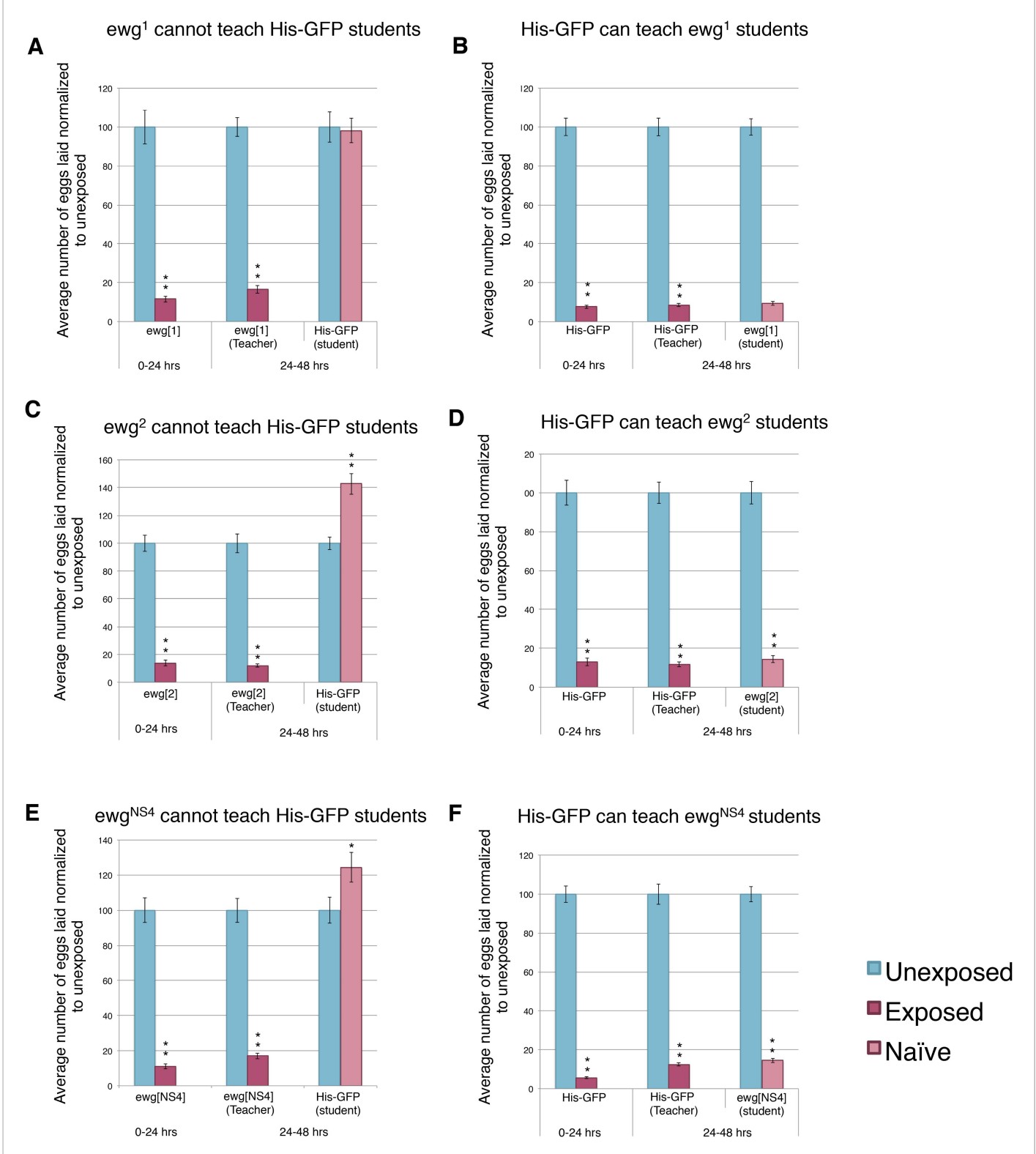

**Figure 11**. Teacher–student dynamics require functional wings to allow for communication to take place. (For **A** to **F**) Percent of eggs laid normalized to unexposed. (**A** and **B**) ewg[1] as teachers and students. (**C** and **D**) ewg[2] as teachers and students. (**E** and **F**) ewg[NS4] as teachers and students. Error bars represent standard error (For [**A** to **F**] $n = 24$ biological replicates.) (*p < 0.05, **p < 1.0e-5).

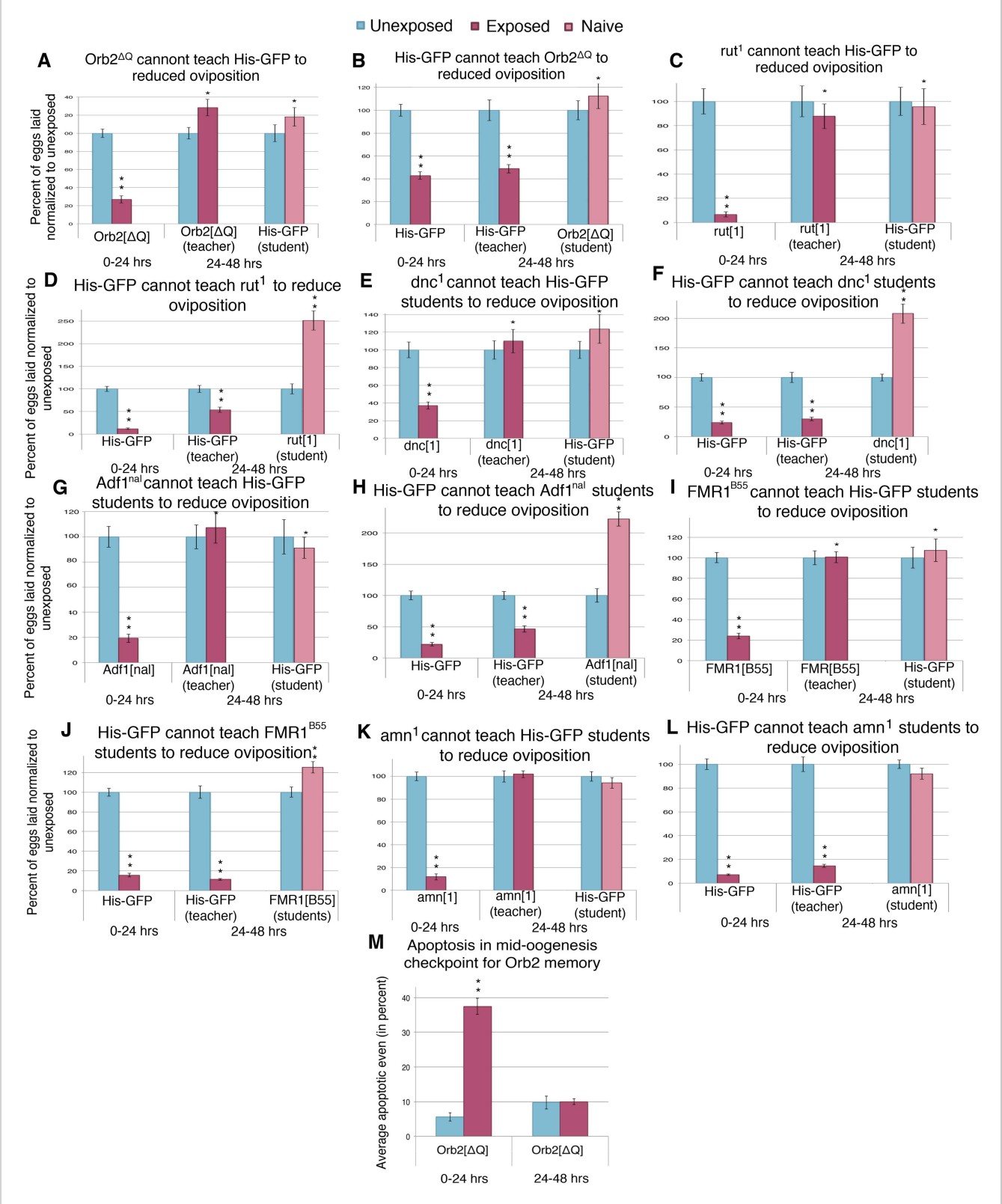

**Figure 12**. Learning mutants are unable to teach or be students. (**A** to **L**) Percent of eggs laid normalized to unexposed. (**A** to **B**) Orb2$^{\Delta Q}$ as teacher and student. (**C** to **D**) $rut^1$ as teacher and student. (**E** and **F**) $dnc^1$ as teacher and student. (**G** and **H**) $Adf1^{nal}$ as teacher and student. (**I** and **J**) $FMR1^{B55}$ as teacher and student. (**K** and **L**) $amn^1$ as teacher and student. For (**M**), average percent of apoptotic events for stage 7/8 egg chambers. (**M**) Orb2$^{\Delta Q}$ exposed and

*Figure 12. continued on next page*

*Figure 12. Continued*

unexposed ovary apoptosis. Error bars represent standard error. (For [**A**] to [**L**] n = 24 biological replicates.) (For [**M**] n = 3 biological replicates from which 12 ovaries were scored for each group) (*p < 0.05, **p < 1.0e-5).

The following figure supplement is available for figure 12:

**Figure supplement 1**. Learning mutants are unable to teach or be students.

as before: in this case, the MB driver (*OK-107-GAL4*) drove expression of an RNA-hairpin targeting *orb2* mRNA. We found that RNAi depletion of Orb2 in the MB produced the same phenotype as the *orb2*$^{\Delta Q}$ mutant tested (*Figure 12A,B*, *Figure 14A–B*). This result highlights that flies deficient in *orb2* in the MB are able to perceive and respond to wasps, but not remember exposure, and therefore cannot teach naive students, once wasps are removed. Flies deficient in *orb2* in the MB are also unable to learn from wild-type teachers. Control parental lines with either just the *OK-107-GAL4* or UAS-Orb2-hairpin transgenes (but not both) functioned as wild type as they exhibited no defects in behavior persistence (*Figure 13—figure supplement 1A–D*, *Figure 14—figure supplement 1A,B*). Control lines expressing RNA-hairpin targeting the white gene in the MB demonstrated wild-type behavior, demonstrating induction of the RNA-hairpin alone does not induce deficient memory formation, teaching ability, or learning ability (*Figure 14C,D*, *Figure 14—figure supplement 1C,D*). This suggests that *orb2* is required in MB neuronal circuits in order for maintained wasp-induced oviposition depression, and it further suggests that persistence of this behavior likely requires long-term memory formation is the MB.

The above data, however, do distinguish between two possible roles for *orb2*. First, the *orb2* gene product could be required for normal development of the MB and other parts of the nervous system that interface with the MB. The *OK-107-GAL4* driver begins expression of GAL4 in the larvae. Thus, it remains possible that RNAi depletion of Orb2 in the larvae could cause developmental defects that then indirectly cause behavioral phenotypes in adults. A second possibility is that persistence of depressed oviposition and in turn teaching ability requires *orb2* function in the adult MB, regardless of

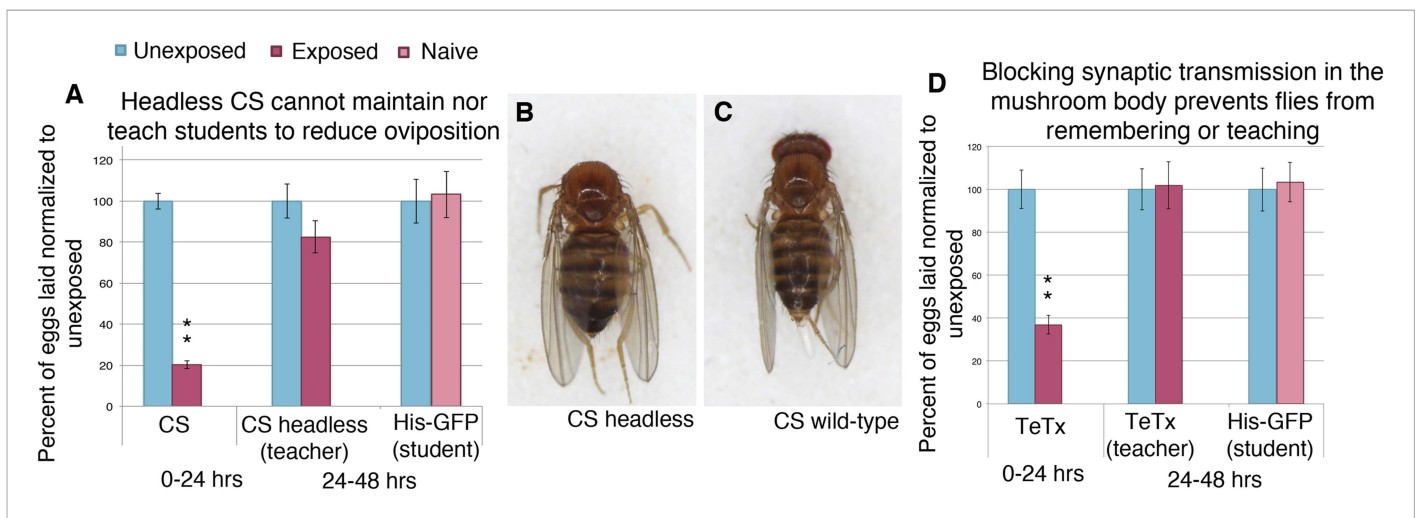

**Figure 13**. Learning and teaching require a continuous neural input from the brain. (**A** and **D**) Percent of eggs laid normalized to unexposed. (**A**) Canton-S teachers with heads removed after acute exposure. (**B**) Dorsal view of representative Canton-S female. (**C**) Dorsal view of representative Canton-S female with no head. (**D**) Flies expressing tetanus toxin (*UAS-TeTx*) in mushroom body (MB) as teacher. Error bars represent standard error (For [**A**] and [**D**] n = 24 biological replicates.) (**p < 1.0e-5).

The following figure supplement is available for figure 13:

**Figure supplement 1**. Blocking synaptic transmission in the MB prevents teacher behavior and student learning.

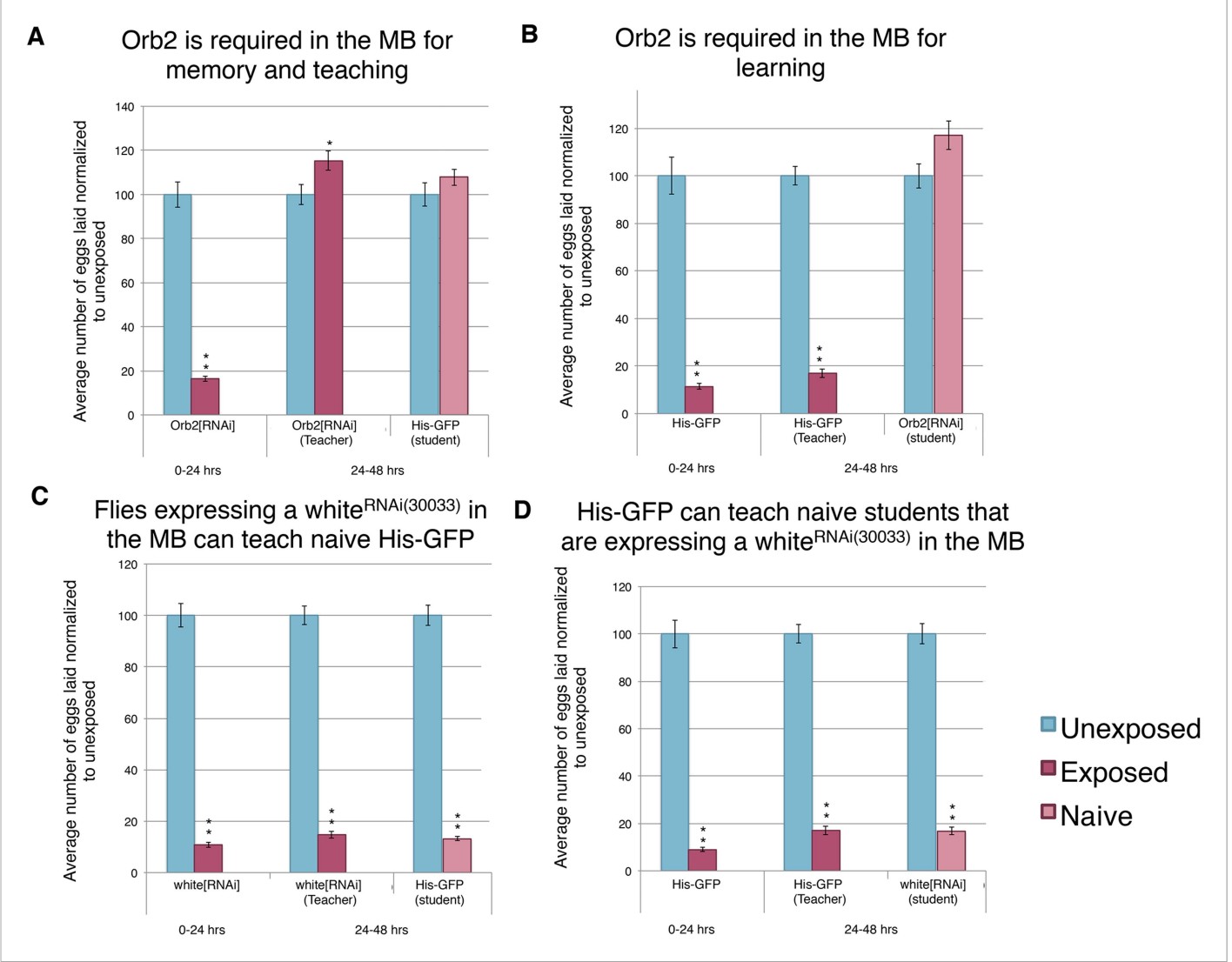

**Figure 14**. Knockdown of Orb2 in the MB results in defective learning. (**A** to **D**) Percent of eggs laid normalized to unexposed. (**A** to **B**) Orb2 RNAi-knockdown as teachers and students. (**C** to **D**) white RNAi-knockdown as teachers and students. (For [**A**] to [**D**] n = 24 biological replicates.) (*p < 0.05, **p < 1.0e-5).

The following figure supplement is available for figure 14:

**Figure supplement 1**. Expression of an RNAi hairpin in the MB does not induce defective learning and memory.

its possible function during MB development. In order to address this question, we turned to the GAL4-based Gene-Switch System where the GAL4 transcription factor is fused to the human progesterone ligand-binding domain (*Burcin et al., 1999*). We used flies expressing the Gene-Switch transgene specifically in the MB, where only an administration of the pharmacological Gene-Switch ligand RU486 could activate the GAL4 transcription factor (*Mao et al., 2004*). In order to confirm our feeding protocol could work in The Fly Condo, we used the MB Gene-Switch line to express a nuclear-localized GFP. Flies were placed into condos containing instant *Drosophila* media hydrated by a mixture of RU486 dissolved in methanol and water. We found that flies placed in the Fly Condo where the food contains RU486 are able to function as wild-type teachers and students (*Figure 15A–B*). This observation demonstrates that RU486 does not perturb *Drosophila*'s ability to perceive and respond to wasp presence by changing their oviposition behavior, as both flies expressing a Gene-Switch construct and His-GFP flies behaved as wild type. Our data also demonstrate that induction of

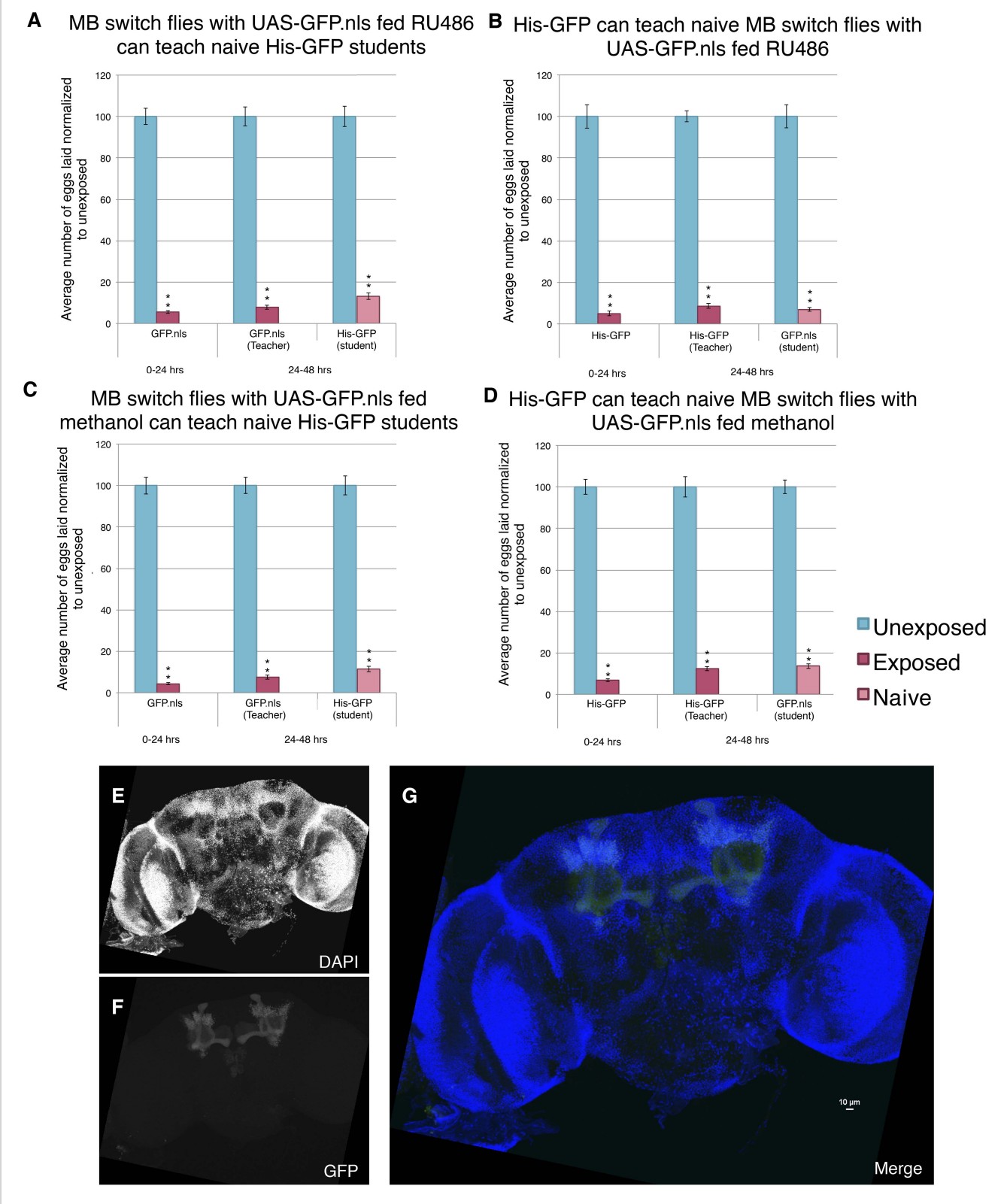

**Figure 15**. Induction of GFP in the MB using the Gene-Switch System does not perturb learning and memory. (**A** to **D**) Percent of eggs laid normalized to unexposed. (**A** to **B**) GFP induction with RU486 feeding in the MB as teachers and students. (**C** to **D**) Lack GFP induction with methanol feeding in the MB

*Figure 15. continued on next page*

Figure 15. Continued
as teachers and students. Brains from flies expressing the GeneSwitch construct (RU486+) in the MB along with a GFP nuclear localization signal (nls) showing (E) DAPI, (F) GFP expression, and (G) the merged image. Scale bar = 10 μm. (For [A] to [D] n = 24 biological replicates.) (**p < 1.0e-5).
The following figure supplement is available for figure 15:

Figure supplement 1. Further evidence demonstrating that induction of GFP in the MB using the GeneSwitch System does not perturb learning and memory.

a protein in MB, in this case GFP, does not perturb learning and memory formation nor teaching ability. When the assay is run with just methanol, therefore lacking RU846, we find flies are able to function as wild type, similar to when they were fed RU486 (*Figure 15C–D*). Control MB-Gene switch parental lines behave as wild-type flies as both homozygotes and when outcrossed to *Canton-S*. In cases when RU486 laden food was fed to flies containing the MB Gene Switch and GFP nuclear localization signal (nls) construct, we find that 24 hr is sufficient to induce GFP signal specifically localized to the MB, whereas food lacking RU486 (methanol only) does not induce GFP after 24 hr (*Figure 15E–G*, *Figure 15—figure supplement 1E–G*).

Given the successful feeding protocol and the MB Gene-Switch construct specificity, we used the MB Gene-Switch to express an RNA-hairpin targeting mRNA for Orb2. Induction of the RNA-hairpin through RU486 feeding in the MB was expected to occur within the same window of time as the GFP expression (*Figure 15*). Flies expressing the MB Gene-Switch and carrying the UAS-Orb2-RNA-hairpin construct, that were not fed RU486, showed normal, wild-type memory, learning, and teaching ability (*Figure 16C–D*). Flies expressing the MB Gene-Switch and carrying the UAS-Orb2-RNA-hairpin construct, which *were* fed RU486, showed a wild-type acute response, but impaired memory formation, learning, and teaching abilities (*Figure 16A,B*). These two data points suggest that the UAS-Orb2-RNA-hairpin construct is only driven in flies expressing the MB Gene-Switch when fed RU486 only. When the MB Gene-Switch parental control line was used to express an RNA-hairpin to the white gene, flies elicited wild-type memory formation with and without RU486 feeding, demonstrating that the Gene-Switch ligand (RU486) alone and an RNA-hairpin alone is not responsible for memory, teaching, and learning impairment (*Figure 16E–H*, *Figure 16—figure supplement 1A–D*). This observation again demonstrates that RU486 does not perturb *Drosophila*'s ability to perceive and respond to wasp presence and that *orb2* function is required for formation of a long-term memory of wasp exposure and not perception of and an acute response to wasps.

Collectively, these data indicate that normal *orb2* function is required in the adult MB for normal long-term memory formation and behavioral changes that persist over multiple days, such as the ability to teach. Use of the MB Gene-Switch construct provides strong evidence to delimit temporal and spatial expression requirements for *orb2* function in the context of this memory assay. Importantly, Orb2-RNAi knockdown in the MB using either *OK107-GAL4* or MB Gene-Switch did not prevent oviposition depression to occur when flies were in the presence of wasps. This also demonstrates that loss/diminution of *orb2* function in the MB does not affect perception and acute response to this predator (*Figure 14A,B*, *Figure 16A–B*).

## Discussion

In this study we have shown that *Drosophila* exhibit an acute response to predatory wasp that entails apoptosis of germ line cells within the ovary and corresponding reduced egg-laying behavior. The response persists over multiple days when learning and memory functions are intact. We also find that this behavior can be socially transmitted from experienced teacher females to naive student females: the transfer of information from teachers does not occur as a by-product of apoptosis in the teacher, but rather through an independent pathway, since depressed oviposition is not a necessary condition for social transmission of reduced egg-laying behavior or apoptosis in the student females (*Figure 17*). These conclusions are further supported by the unexpected observation that student flies, that had learned to reduce oviposition, could not serve as teachers (*Figure 2B*, *Figure 2—figure supplement 1*). We emphasize that teacher-instructed students continued to exhibit depressed oviposition and stage 7/8 egg chamber apoptosis in the 24-hr period after removal of teachers. This again indicates that depressed oviposition itself is not

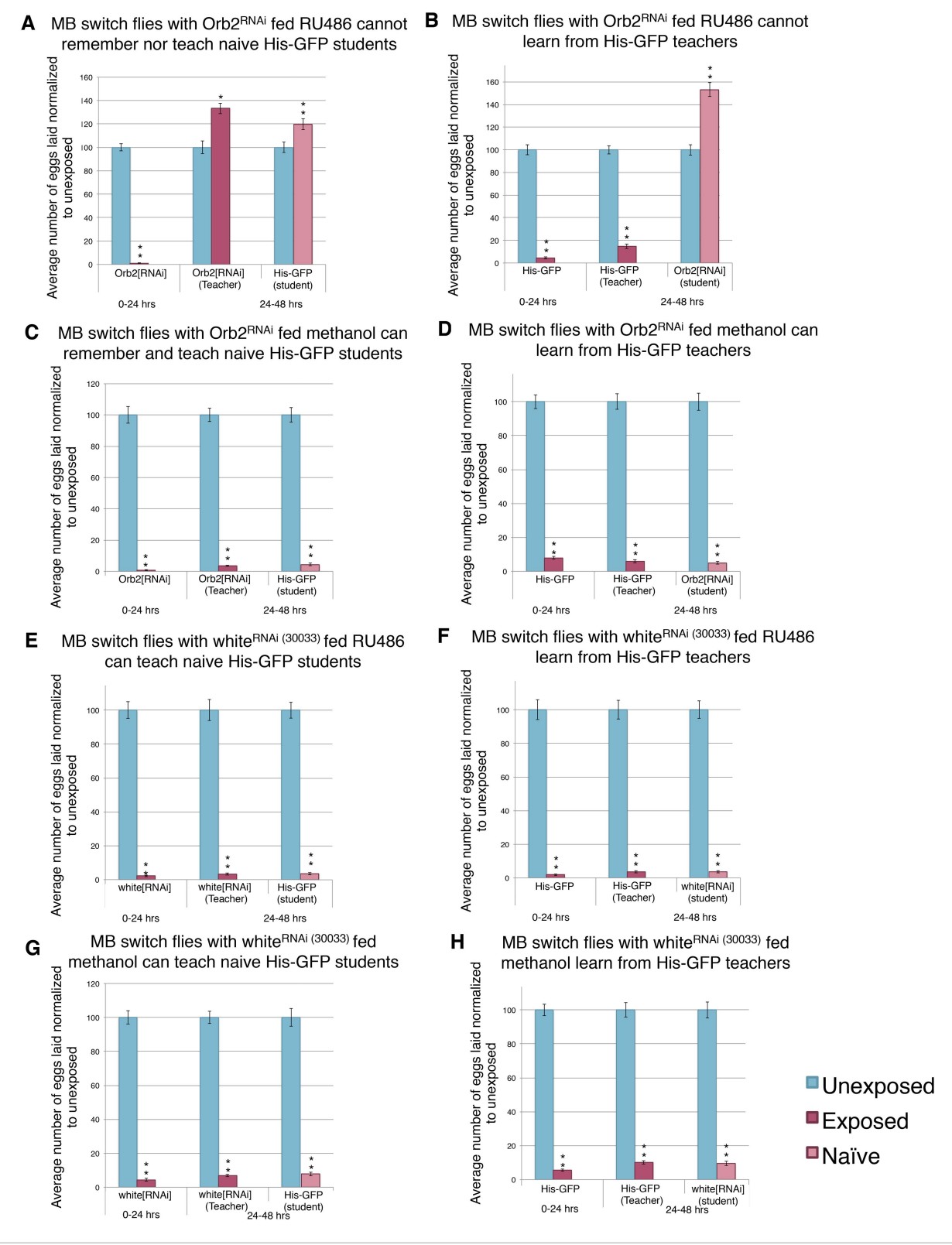

**Figure 16.** Knockdown of Orb2 in the MB using the GeneSwitch System results in defective learning. (**A** to **H**) Percent of eggs laid normalized to unexposed. (**A** to **B**) Orb2 RNAi-knockdown in the MB (GeneSwitch) fed RU486 as teachers and students. (**C** to **D**) Orb2 RNAi-knockdown in the MB

*Figure 16. continued on next page*

*Figure 16. Continued*

(GeneSwitch) not fed RU486 (methanol fed). (**E** to **F**) White RNAi-knockdown in the MB (GeneSwitch) fed RU486 as teachers and students. (**G** to **H**) White RNAi-knockdown in the MB (GeneSwitch) not fed RU486 (methanol fed). (For [**A**] to [**H**] n = 24 biological replicates.) (*p < 0.05, **p < 1.0e-5).

The following figure supplement is available for figure 16:

**Figure supplement 1**. Expression of an RNAi hairpin in the MB using the GeneSwitch System does not perturb learning and memory.

sufficient for information transfer. However, at a higher level, these observations also indicate that such adaptive information transfer cannot spread throughout a population, since only primary teachers are able to transmit the predator-threat information.

The above findings document a pathway initiated through visual stimulation and results eventually in a dramatic physiological response in the ovary. The discovery of neurally driven control of non-neural germ line cell physiology is conceptually similar to a recent study in *Drosophila*, which demonstrated that olfactory stimulation was necessary for maintenance of blood progenitor cells (*Shim et al., 2013*), thus, also establishing a link between perception of environmental information and physiological response to specific information. Although learning mutants and flies expressing an RNA-hairpin to orb2 could perceive and respond to predator presence, the observation that egg production completely recovered by 24 hr following removal of the wasp threat (*Figure 12*, *Figure 12—figure supplement 1*, *Supplementary file 1K,L*) is consistent with previous observations where females switched from a poor to rich food source repress the mid-oogenesis checkpoint via insulin signaling and recover normal egg production within 24 hr (*Drummond-Barbosa and Spradling, 2001*). This rapid recovery of oviposition in learning and memory mutants, coupled with removing fly heads and inhibiting synaptic transmission in the MB, suggests that maintenance of the depressed oviposition state requires continued neural signaling mediated by a memory component of the brain.

Our observations document and describe a particularly robust form of social learning in *Drosophila* and establish several fundamental features. First, direct learning and social learning require visual system function but occur through different mechanisms: in particular, the acute response of flies to direct wasp exposure can occur even in classic-learning mutants, while persistence of the predator response and subsequent social learning requires functions of learning genes and continued neural input. Loss of memory gene functions, such as *Adf1, amn, dnc, dFmr1, rut*, and *Orb2*, or inhibition of MB synaptic transmission had no effect on the ability to change oviposition behavior in the presence of wasp, however, in each of these cases, persistence of this behavior after wasp removal, and subsequent teaching ability, was abolished. Additionally, inhibition of orb2 using the GAL4/UAS and Gene-Switch systems suggests that maintenance of the change in oviposition state requires neural signaling mediated by a memory component of the adult brain. Second, social learning occurs through a mechanism distinct from mimicry. Information of wasp presence can be transmitted by animals that have encountered wasps but are physiologically unable to display egg retention, which is the normal behavioral output of such learning (*Figure 17*). Third, social learning in this context appears to be limited in its spread: being transmitted only from teachers with direct predator experience to students that they encounter. Therefore, students that have learned through social learning cannot become teachers themselves (*Figure 2*). This is noteworthy because the inability of primary students to further transfer information to secondary students will limit the time frame and number of individuals in which this knowledge transfer takes place. The spreading of socially learned behavior has been previously postulated to possibly drive local adaptation by maintaining behavioral diversity of groups through self-propagating social learning once initiated in an individual (*Battesti et al., 2012*). With regards to social learning of oviposition depression in response to a predator threat, it seems reasonable that such information would be most useful if limited to nearest neighbors, whose progeny may be similarly vulnerable in time and space by parasitoid wasps. However, the fitness costs of prolonged oviposition depression and/or spreading to conspecifics beyond primary learners could be devastating if it were self propagating, and thus, the degree to which it can spread within a group must be limited by restricting teaching behavior only to individuals having had direct visual experience of the threat, while ensuring memory of the threat in both primary (teachers) and secondary (students) learners is maintained and then decays over time.

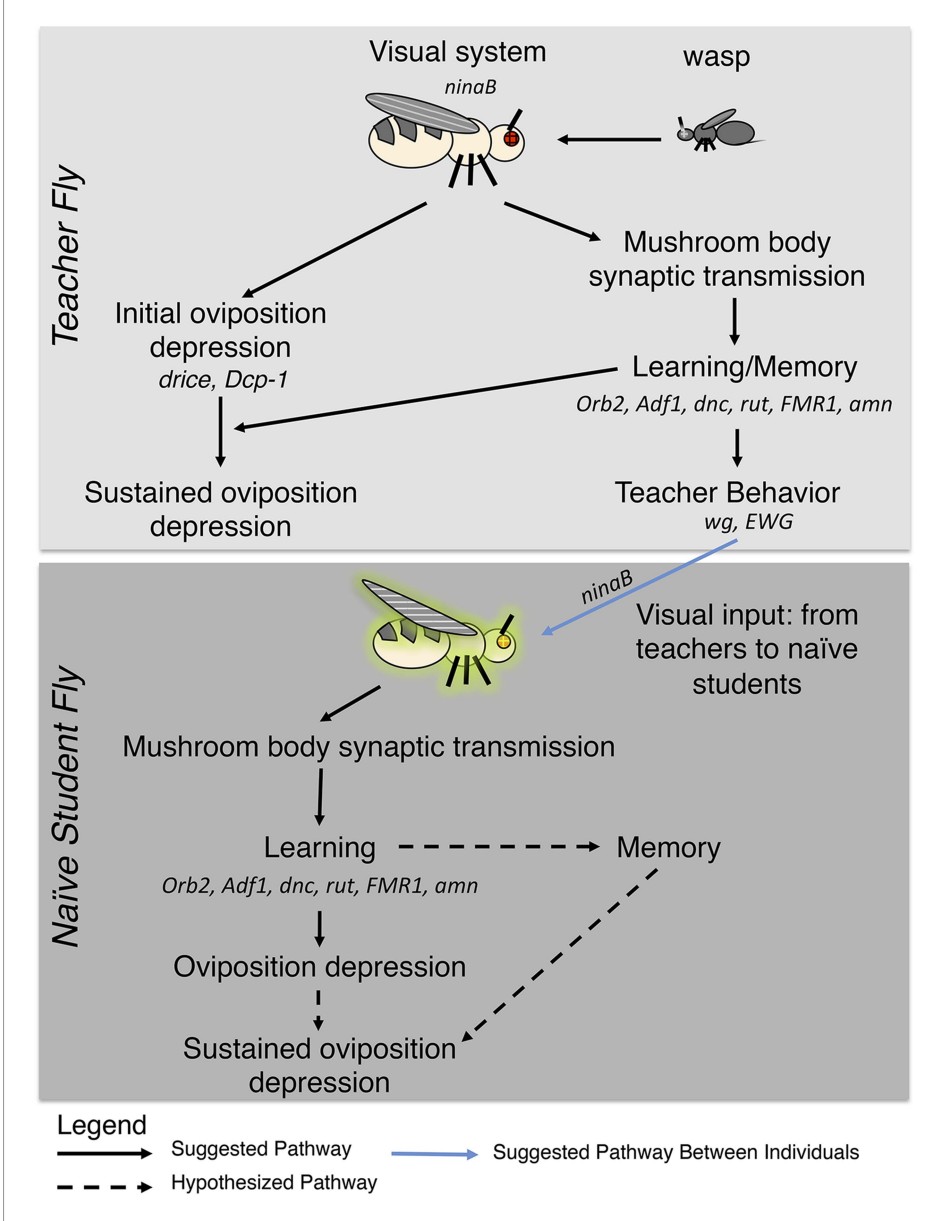

**Figure 17**. Pathway model for fly-wasp mediated social learning. Initial oviposition depression during the 0- to 24-hr acute response period and information transmission during social learning 24- to 48-hr periods are not coupled. Sustained oviposition depression requires learning and memory genes in both teachers and students. Alleles tested for indicated genes were $ninaB^{P315}$, $Orb2^{\Delta Q}$, $Adf1^{nal}$, $dnc^1$, $dnc^{ML}$, $rut^1$, $rut^{2080}$, $FMR1^{B55}$, $FMR1^3$, $amn^1$, $amn^{X8}$, $wg^1$, $ewg^1$, $ewg^2$, $ewg^{Ns4}$, and drice-RNAi, Dcp-1-RNAi, $Dcp-1^1$, $Dcp-1^3$.

In sum, we have shown that visual inputs modify synaptic signaling in the MB of the fly brain to implement a behavioral and physiological change, both of which are transferable through extrinsic inputs to naive student flies, and that experiments based around wasp exposure can serve as a simple and robust learning, social learning, and memory paradigm in future *D. melanogaster* studies. The learning and memory genes we tested and found to be involved are conserved across many animal species (***Bolduc and Tully, 2009***), and thus, serves as an excellent approach to model cellular and neuronal network functions that may be relevant to vertebrate brain function. Even though the vertebrate brain is vastly more complex than that of the fly, additional genes, gene families, and pharmacological effects can be elucidated in *Drosophila* and may identify core mechanisms that are

used in all species. These conserved components provide starting points in vertebrate animals for further vertical integration in the fields of learning and social communication. In this way, mechanisms that are unique to vertebrates can also be inferred, and we suggest that the learning and memory paradigm presented here will prove to be a useful discovery tool. We believe this study establishes a new and robust ecologically relevant model of social learning in *Drosophila* with possible far reaching implications for neurobiology, Darwinian selection and evolution.

## Materials and methods

### Insect species/strains

The *D. melanogaster* strains Canton-S (CS), Oregon-R (OR), $w^{1118}$, and transgenic flies carrying Histone H2AvD-GFP (His-GFP) were used as wild-type strains for assaying egg retention in the presence of wasps. All subsequent experiments we performed using either CS or His-GFP flies as wild type. $Orco^1$, $ninaB^{P315}$, $Dcp-1^{RNAi}$, $drice^{RNAi}$, Histone H2AvD-GFP, and the *Matα* GAL4 mutant strains were acquired from the Bloomington *Drosophila* Stock Center (strain numbers 23129, 24776, 28909, 32403, 35518, and 7063, respectively). $dnc^1$, $dnc^{ML}$, $rut^1$, $rut^{2080}$, $amn^1$, and $amn^{X8}$ were kindly provided by Leslie Griffith (Brandeis University). $Wg^1$, $ewg^1$, $ewg^2$, $ewg^{NS4}$, MS1096 GAL4, and UAS-Reaper (UAS-Rpr) flies were kindly provided by Yashi Ahmed (Geisel School of Medicine at Dartmouth). $Dcp-1^2$ and $Dcp-1^3$ were kindly provided by Kim McCall (Boston University). The MB Gene-Switch line and the MB-247 were kindly provided by Greg Roman (Baylor College of Medicine) (*Supplementary file 2*). All flies were maintained on standard cornmeal/yeast/molasses *Drosophila* medium. For all outcrosses, *Canton-S* virgin females were mated to males of the appropriate genotype.

Flies aged 3–5 days post-eclosion on fresh, molasses-based, *Drosophila* media were used in all experiments. Stocks were maintained at 25°C in 70% humidity with a 12:12 light:dark cycle. For stocks maintained in vials, 25 females were kept for stocks with 10 males at maximum to prevent over-crowding. Stocks kept in bottles had a maximum of 100 females and 40 males to prevent over-crowding. When flies were close to eclosion, parents were removed from the bottles. Newly eclosed flies were moved to fresh *Drosophila* media (in bottles or vials at the same population density) and aged until they were between 3 and 5 days of age maintained at 25°C in 70% humidity with a 12:12 light:dark cycle, at which point they were used in experiments. We stress the importance of aging the flies on fresh media, as it appears that flies aged on old media (i.e., the same media in which they eclosed) are nutrient deprived and naturally lay very few eggs.

The Figitid larval endoparasitoid *Leptopilina heterotoma* (strain Lh14) was used in all experiments. *L. heterotoma* strain Lh14 originated from single females collected in Winters, California in 2002, and was kindly provided by Todd Schlenke (*Schlenke et al., 2007*). In order to culture wasps, adult flies were allowed to lay eggs in standard *Drosophila* vials containing standard *Drosophila* medium for 4 days before being replaced by adult wasps (10 female, 6 male), which then attack the developing fly larvae. Wasp vials were supplemented with approximately 500 µl of a 50% honey/water solution applied to the inside of the cotton vial plugs. Wasps aged 3–7 days post-eclosion were used for all experiments. Fresh wasps were used for all experiments, such that wasps were never reused between experiments.

### Fly oviposition

Fly oviposition rates were conducted using The Fly Condo (Genesee Scientific (San Diego, CA) Cat # 59-110) (*Figure 1A*), which contained 24 independent chambers. Each chamber is 7.5 cm long by 1.5-cm diameter. Each condo/chamber had a bottom 24-well food plate with approximately 2 ml of standard, molasses cornmeal media per chamber. Briefly, bottles containing *Drosophila* were microwaved for 30 s at maximum heat. This liquid food was allowed to cool before dispensing 2 ml into the Fly Condo plates, where food was allowed to cool for another 30 min before the start of the experiment. All experiments used this food protocol unless otherwise noted (specifically experiments using instant *Drosophila* media with RU486 experiments). Mesh wire was along the top of the condo, allowing air transfer. In order to assay egg retention of flies in the presence of wasps (acute exposure), 5 female flies and 1 male fly (prepared and aged as described above) were placed into one chamber of The Fly Condo in the control, while 3 female Lh14 wasps were placed with the flies in the experimental setting. The oviposition plate from control and experimental condos was made 24 hr later.

In order to assay fly communication and the social learning period, 5 female flies and 1 male fly were placed into one chamber of The Fly Condo in the control, while 3 female Lh14 wasps were placed

with the flies in the experimental setting for 24 hr. After the 24-hr exposure, wasps were removed by anesthetizing flies and wasps in the condos. Control flies underwent the same anesthetization. Wasps were removed and replaced with 3 female 'student' flies. All flies were placed into new clean condos for the second 24-hr period. The oviposition plate from each fly condo was replaced 24 hr after the start of the experiment, and the second plate was removed 48 hr after the start of the experiment. Fly egg counts from each plate were made at the 0–24-and 24–48-hr time points. To control for both seasonal influence and population effects of both flies and wasps used, acute- and social-learning period experiments were repeated in 24-experimental replicate increments in both August 2013 and April 2014. We found the same effect in both time points tested, suggesting that seasonal changes and population effects were not affecting our results (*Figure 1C* and *Figure 1—figure supplement 1C–E*).

In order to demonstrate that students cannot become teachers, the same protocol as above was performed (*Figure 2*). At the 48-hr time point, exposed teacher flies were removed by anesthetizing teacher and 1° student flies in the condos. Control flies underwent the same anesthetization. The exposed teacher flies were replaced with 3 new naive student flies, termed 2° students. These 2° students were placed with 1° student flies into new condos for the third 24-hr period. Fly egg counts from each oviposition plate were made at the 0–24, 24–48, and 48–72 hr time points.

In order to demonstrate that teachers could teach more than one cohort of students, the same protocol as above was performed with the exception of teacher removal (*Figure 3*). At the 48-hr time point, 1° student flies were removed by anesthetizing students and teachers and teacher flies were placed into new condos. The 1° student flies were replaced with 3 new naive student flies, termed 2° students. These 2° students were placed with teacher flies into new condos for the third 24-hr period. Fly egg counts from each oviposition plate were made at the 0–24, 24–48, and 48–72 hr time points.

To assay if the ratio of teachers to students impacted the ability for information transfer, 3 female flies were placed into one chamber of The Fly Condo in the control, while 3 female Lh14 wasps were placed with the flies in the experimental setting for 24 hr. Wasps were then removed and replaced with 3 female, naive student flies of the opposite genotype (either His-GFP or *Canton-S* depending on teacher identity). Flies were then placed into new, clean condos. This provided a 1:1 ratio of teachers to students. The same protocol was performed to see if males participated in transmission of information by having 3 male flies exposed to 3 female Lh14 wasps. Wasps were then removed and replaced with 3 female, naive student flies of the opposite genotype (either His-GFP or *Canton-S* depending on teacher identity).

In order to assay the role of light during initial exposure and the role of light during the learned response, multiple assays were performed where light availability was varied (*Figure 8*). For experiments where the acute response occurred in the dark, flies were anesthetized and placed into The Fly Condo with or without wasps as described above. However, they were then immediately placed into a box, taped closed with Duct Tape (to prevent light leaks), and allowed to awaken in the dark. Flies were kept in the dark for either 24 hr, after which wasps were removed and students were added and moved into the light, or kept in the dark for the duration of the experiment (48 hr) including the social-learning period with students. If flies were to be kept in the dark, the only light they were exposed to was just before they were anesthetized and given students.

All treatments were run at 25°C at 70% humidity with a 12:12 light:dark cycle in twenty-four replicates unless otherwise noted with both teacher and student flies aged 3–5 days post-eclosion. Food used for Fly Condo plates was the same molasses based *Drosophila* media used in maintaining fly stocks, unless otherwise noted. Fly condos and oviposition plates were bleached thoroughly with 10% bleach and rinsed with distilled water mixed with Sparkleen after every use (1 gallon of water: 1 gram of Sparkleen). All egg plates were coded and scoring was blind as the individual counting eggs were not aware of treatments or genotypes.

To assay whether flies continued to eat high-nutrient food during wasp exposure, flies were placed into a large embryo collection chamber (Genesee Scientific (San Diego, CA) Cat No. 59-101), which fits a 100-mm Petri dish. Dishes were filled with 5 grams of blue instant drosophila media (Fisher Scientific (Pittsburgh, PA) Cat No. S22315C), supplemented with a total of 20 ml of distilled water to hydrate the food. Yeast paste was made with 15 ml distilled water, 5 mL McCormick's red food dye, and 13 mL live yeast. Approximately, 15 mL of the yeast paste solution was added to the center of the petri dish containing the instant *Drosophila* media. In the egg lay chambers, 100 female *Canto- S* and 20 male *Canton-S* flies were added for control conditions. For exposed

conditions, 100 female *Canton-S* and 20 male *Canton-S* flies were added with the addition of 50 female Lh14. The experiment was run for 24 hr at 25°C in 70% humidity on a 12:12 light:dark cycle. After 24 hr, flies were anesthetized and were scored for color in their abdomens. A random subset of 12 females was taken after abdominal quantification for ovary dissection and DAPI staining. Three replicates were performed for these experiments.

## Mechanical manipulation

To assay whether or not wings were involved in the information transmission in the social learning period, 5 female and 1 male *Canton S* were anesthetized and their wings were cut at the base using micro-scissors (Fine Science Tools (Foster City, CA); Item No. 15,001-08). Following clipping, flies were placed into one chamber of The Fly Condo in the control, while 3 female Lh14 wasps were placed with the flies in the experimental setting for 24 hr. After the 24-hr exposure, wasps were removed by anesthetizing flies and wasps in the condos. Control flies underwent the same anesthetization. Wasps were removed and replaced with 3 female 'student' flies. All flies were placed into new clean condos for the second 24-hr period. The oviposition plate from each fly condo was replaced 24 hr after the start of the experiment, and the second plate was removed 48 hr after the start of the experiment. Fly egg counts from each plate were made at the 0–24 and 24–48 hr time points.

In order to assay whether a continued input from the brain is needed for flies to remember wasp exposure and to transmit that information, 5 female flies and 1 male fly were placed into one chamber of The Fly Condo in the control, while 3 female Lh14 wasps were placed with the flies in the experimental setting for 24 hr. After the 24-hr exposure, wasps were removed by anesthetizing flies and wasps in the condos. Control flies underwent the same anesthetization. During this anesthetization period, both male and female flies were decapitated using the micro-scissors. Decapitated flies that were not standing after anesthesia recovery were excluded. Wasps were removed and replaced with 3 female 'student' flies. All flies were placed into new clean condos for the second 24-hr period. The oviposition plate from each fly condo was replaced 24 hr after the start of the experiment, and the second plate was removed 48 hr after the start of the experiment. Fly egg counts from each plate were made at the 0–24 and 24–48 hr time points.

## Fly duplexes

Fly duplexes (*Figure 9*) were constructed by using three standard 25 mm × 75-mm glass microscope slides (VWR (Radnor, PA): Item No. 48,300-025) that were adhered between two 75 mm × 50 mm × 1-mm glass microscope slides (Fisher: Item No. 12-550C). Clear aquarium silicone sealant was used to glue these glass slides together, making two compartments separated by one 1-mm thick glass slide. Sealant was allowed to cure for 48 hr; each duplex was then soaked in water and Sparkleen detergent overnight (1 gallon distilled water: 1 gram Sparkleen), rinsed in distilled water (dH$_2$O) overnight, rinsed with 70% ethanol and air-dried. The interior dimensions of each of the two units measured approximately 23.5 mm (wide) × 25 mm (deep) × 75 mm (tall).

For experiments using Fly Duplexes, plates from The Fly Condo (Genesse Cat. Item No. 59-113) were filled to the rim with standard *Drosophila* media and allowed to cool. Upon cooling, a single Fly Duplex was inserted into the food such that it touched the bottom of the plate. The open end of the Fly Duplex was closed using a cotton plug (Genesse Scientific (San Diego, CA) Cat. Item No. 51-102B) to prevent insect escape. 10 female flies and 2 male flies were placed into one chamber of the Fly Duplex in the control, while 10 female Lh14 wasps were placed in the compartment adjacent to the flies in the experimental setting for 24 hr. After the 24-hr exposure, flies and wasps were removed by anesthetizing flies and wasps in the Fly Duplexes. Control flies underwent the same anesthetization. Wasps were removed and replaced with 10 female 'student' flies. All flies were placed into new clean Duplexes for the second 24-hr period. The oviposition plate from each fly condo was replaced 24 hr after the start of the experiment, and the second plate was removed 48 hr after the start of the experiment. Fly egg counts from each plate were made at the 0–24 and 24–48 hr time points.

## RU486 feeding

RU486 (Mifepristone) was used from Sigma-Aldrich Corp. (St. Louis, MO) (Lot Item No. SLBG0210V). Condos were prepared by measuring 0.375 grams of flaky instant blue *Drosophila* medium into each well of The Fly Condo plates. For all food treatments, we pipetted a total liquid volume of 2250 µl directly onto the instant food. For experiments with RU486, an RU486 solution was used. This was

prepared by dissolving 3.575 mg of RU486 in 800 µl methanol (Fisher Scientific (Pittsburgh, PA) Lot number 141313). This solution was added to 15.2 ml of distilled water. The total solution (16 ml) was thoroughly mixed and 2250 µl were pipetted onto the instant food into each well. For plates containing no RU486 (methanol only), 800 µl methanol was mixed with 15.2 ml of distilled water. The total solution (16 ml) was thoroughly mixed and 2250 µl were pipetted onto the instant food into each well.

## Immunofluorescence

Ovaries that were prepared for immunofluorescence were fixed in 4% methanol-free formaldehyde in PBS with 0.001% Triton-X for approximately 5 min. The samples were then washed in PBS with 0.1% Triton-X and blocked with 2% normal goat serum (NGS) for 2 hr. The primary antibody, rabbit cleaved caspase-3 (Cell Signaling (Beverly, Massachusetts) 5 A1E) at a concentration of 1:400, was incubated overnight at 4°C in 2% NGS. The secondary antibody used was Cy3 conjugated (Jackson Immunoresearch (West Grove, PA)) and used at a concentration of 1:150 during a 2-hr incubation at room temperature. This was followed by a 10-min nuclear stain by DAPI.

In order to assay whether feeding flies RU486 in The Fly Condo would be sufficient to turn on the MB gene switch construct, we placed flies into condos containing RU846$^+$ food. Flies had the MB switch construct as well as a UAS-GFP nls construct, such that if the MB switch is activated, it should fluoresce with GFP. After a 24-hr period in The Fly Condo, adults were removed and fixed in 4% methanol-free formaldehyde in PBS with 0.001% Triton X overnight at 4°C. Brains were then dissected out of whole adults in PBS. The samples were then washed in PBS with 0.1% Triton X and stained with DNA staining with DAPI, for 10 min and mounted in Vectashield (Vector Laboratories (Burlingame, CA) Item No. H-1000) before imaging.

## TUNEL

Individual ovarioles were dissected and fixed in PBS with 4% methanol-free formaldehyde and 0.1% Triton-X for 30 min. Ovarioles were washed and incubated in PBS with 20 µg/ml proteinase K for 10 min. Recombinant terminal transferase (Tdt) labeling was conducted with the use of Cy3-conjugated dUTP (GE Healthcare (Troy, NY) PA53032). Tdt reaction mixture (200 mM NaCacodylate, 0.1 mM DTT, 1 mM CoCl$_2$, 0.05 mM Cy3-dUTP, 0.05 mM dTTP) in Tdt buffer and Tdt enzyme (Roche (Basel, Switzerland) 03333566001) was incubated with samples for 3 hr at 37°C in a dark hybridization oven. At the end of the incubation period, 2 µl of (0.25 M) EDTA was added to stop the reaction. Samples were counter-stained with DAPI, mounted in Vectashield, and stored at −20°C until imaging.

## Apoptosis quantification

For quantification of egg chamber apoptotic events, ovaries from exposed teachers and exposed students (in addition to unexposed controls) were fixed in 4% methanol-free formaldehyde in PBS with 0.001% Triton X for approximately 5 min. The samples were then washed in PBS with 0.1% Triton X and stained with DAPI for 10 min. Batches of student and teacher flies were stained together in the same wells to prevent stain bias. In all cases, student and teacher ovaries on the same slides could be distinguished based on the Histone H2AvD-GFP marker (*Figure 4—figure supplement 1A,B*).

## Imaging

A Nikon (Melville, New York) A1R SI Confocal microscope was used for imaging TUNEL, brain, and caspase staining. Image averaging of 4× during image capture was used for all images unless otherwise specified. A Nikon E800 Epifluorescence microscope with Olympus DP software was used to quantify apoptotic events in egg chambers in addition to the capture of egg images and of whole flies (*Figure 4B,C,F,G*, *Figure 4—figure supplement 1A–J,M–T*). Images of The Fly Condo, oviposition plates with red yeast paste, and low-magnification images of exposed and unexposed flies with red abdomens were made using an iPad 2 operating with ISO 64 (*Figure 1A*, *Figure 4A*, *Figure 4—figure supplement 1K–L*). Images of The Fly Condo and the Fly Duplex were color enhanced in iPhoto (*Figure 1A*, *Figure 9A*).

## Statistical analysis

Statistical tests were preformed in R (version 3.0.2, 'Frisbee Sailing'). Welch's two-tailed t-tests were preformed for all egg count data. p-values reported were calculated for comparisons between paired

treatment-group and unexposed. A chi square test was preformed to determine significance of feeding experiments for frequency of colored abdomens. Welch's two-tailed t-tests were performed on apoptosis data with each exposure batch treated as a replicate (n = 3), in instance where both the treatment and control group had 0% apoptosis across all of the three replicates the p-value was not calculable, and is reported as 'N/A' (See *Supplementary files 3–5*).

## Acknowledgements

We thank Todd Schlenke for initiating one of us (BK) to the experimental system used here and for providing wasp strains. We thank Leslie Griffith, Yashi Ahmed, Kim McCall, Greg Roman, FlyBase, and the Bloomington *Drosophila* Stock Center for stocks, the Dartmouth Department of Biological Sciences Light Microscopy Facility, Rhiannon Greywolf for technical assistance helpful comments on the manuscript, and Huy Nguyen and Heather Wallace for helpful comments on the manuscript. We acknowledge grants from Geisel School of Medicine at Dartmouth and NIGMS K18GM097732 (GB) and Science Foundation Ireland (MR).

## Additional information

### Competing interests

MR: Reviewing editor, *eLife*. The other authors declare that no competing interests exist.

### Funding

| Funder | Grant reference | Author |
|---|---|---|
| National Institute of General Medical Sciences (NIGMS) | | Giovanni Bosco |
| Science Foundation Ireland | | Mani Ramaswami |
| Dartmouth College | Geisel School of Medicine at Dartmouth | Giovanni Bosco |

The funders had no role in study design, data collection and interpretation, or the decision to submit the work for publication.

### Author contributions

BZK, JB, GB, Conception and design, Acquisition of data, Analysis and interpretation of data, Drafting or revising the article; MR, Conception and design, Analysis and interpretation of data, Drafting or revising the article

## Additional files

### Supplementary files

• Supplementary file 1. Absolute number of apoptotic egg chambers. (**A**) Absolute number of apoptotic egg chambers in *Canton-S* flies immediately following wasp exposure or mock exposure (0- to 24-hr period in oviposition experiments). Each replicate, replicate sum, and replicate average along with standard error is shown (*p < 0.05, **p < 0.001). (**B**) Absolute number of apoptotic egg chambers in GFP-Histone flies immediately following wasp exposure or mock exposure (0- to 24-hr period in oviposition experiments). Each replicate, replicate sum, and replicate average along with standard error is shown (*p < 0.05, **p < 0.001). (**C**) Absolute number of apoptotic egg chambers in teacher *Canton-S* flies 24 hr following wasp exposure or mock exposure (24- to 48-hr period in oviposition experiments). Each replicate, replicate sum, and replicate average along with standard error is shown (*p < 0.05, **p < 0.001). (**D**) Absolute number of apoptotic egg chambers in teacher GFP-Histone flies 24 hr following wasp exposure or mock exposure (24- to 48-hr period in oviposition experiments). Each replicate, replicate sum, and replicate average along with standard error is shown (*p < 0.05, **p < 0.001). (**E**) Absolute number of apoptotic egg chambers in *Canton-S* flies fed yeast paste immediately following wasp exposure or mock exposure (0- to 24-hr period in oviposition experiments). Each replicate, replicate sum, and replicate average along with standard error is shown (*p < 0.05, **p < 0.001). (**F**) Absolute number of apoptotic egg chambers in student

GFP-Histone flies 24 hr following wasp exposure or mock exposure (24- to 48-hr period in oviposition experiments). Each replicate, replicate sum, and replicate average along with standard error is shown (*p < 0.05, **p < 0.001). (G) Absolute number of apoptotic egg chambers in student *Canton-S* flies 24 hr following wasp exposure or mock exposure (24- to 48-hr period in oviposition experiments). Each replicate, replicate sum, and replicate average along with standard error is shown (*p < 0.05, **p < 0.001). (H) Absolute number of apoptotic egg chambers in Dcp-1[RNAi] flies immediately following wasp exposure or mock exposure (0- to 24-hr period in oviposition experiments). Each replicate, replicate sum, and replicate average along with standard error is shown (*p < 0.05, **p < 0.001). (I) Absolute number of apoptotic egg chambers in teacher Dcp-1[RNAi] flies 24 hr following wasp exposure or mock exposure (24- to 48-hr period in oviposition experiments). Each replicate, replicate sum, and replicate average along with standard error is shown (*p < 0.05, **p < 0.001). (J) Absolute number of apoptotic egg chambers in student GFP-Histone flies, paired with Dcp-1[RNAi] teachers, 24 hr following wasp exposure or mock exposure (24- to 48-hr period in oviposition experiments). Each replicate, replicate sum, and replicate average along with standard error is shown (*p < 0.05, **p < 0.001). (K) Absolute number of apoptotic egg chambers in $Orb2^{\Delta Q}$ flies immediately following wasp exposure or mock exposure (0- to 24-hr period in oviposition experiments). Each replicate, replicate sum, and replicate average along with standard error is shown (*p < 0.05, **p < 0.001). (L) Absolute number of apoptotic egg chambers in $Orb2^{\Delta Q}$ flies 24 hr following wasp exposure or mock exposure (24- to 48-hr period in oviposition experiments). Each replicate, replicate sum, and replicate average along with standard error is shown (*p < 0.05, **p < 0.001).

• Supplementary file 2. Genotypes of each fly strain used in study. Names used in study, followed by full genotype and location acquired from shown.

• Supplementary file 3. Statistical analyses and corresponding P-values shown for main figures. Comparison groups, statistical test performed, sample size, and P-values are shown for a corresponding figure.

• Supplementary file 4. Corresponding P-values generated from t-tests are shown for supplementary figures. Comparison groups, sample size, and P-values are shown for a corresponding figure.

• Supplementary file 5. Corresponding P-values generated from t-tests are shown for *supplementary file 1A–L*. Comparison groups, sample size, and P-values are shown for a corresponding file.

• Supplementary file 6. Corresponding raw average egg counts corresponding to main figures are shown.

• Supplementary file 7. Corresponding raw average egg counts corresponding to supplementary figures are shown.

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
