## [Decision Letter]

Thank you for sending your work entitled “Predator-induced changes in *Drosophila* behavior and germline physiology are socially communicated by talking flies” for consideration at *eLife*. Your article has been favorably evaluated by Eve Marder (Senior editor) and two reviewers, one of whom, Leslie Griffith, is a member of our Board of Reviewing Editors. Ralph Greenspan has also agreed to share his identity.

This paper provides an important and interesting contribution to the literature: to the evolution literature, for its start at delineating mechanisms for an ethologically-based behavioral response, and to the neurobiological literature as a potent paradigm for learning.

Essential revisions:

1) In a number of places, the data presented in a panel are clearly the wrong data i.e. it is a copy of other panels found in other figures. I noted this in Figure 8—figure supplement 1 and Figure 9, but there may be other instances of this. The authors need to go through the figures very carefully to make sure they are all correct.

2) There are some irregularities in the way genotypes are expressed. “>” usually implies a two-tiered system like GAL4 driving a transgene's expression i.e. two transgenes. In a number of the figures, this symbol is used in a non-canonical way to indicate a single transgene. An example is Figure 13—figure supplement 1 where the labels say “*UAS>TeTx”* when I believe what is trying to be indicated is a *UAS-TeTx/+* genotype. All the figures need to be edited to correct this.

3) Is there information on how flies were reared, aged, stored prior to assays? If I couldn't find it, no problem. If it's not there, it should be.

Note that the title “Predator-induced changes in *Drosophila* behavior and germline physiology are socially communicated by talking flies” needs to be modified. We suggest dropping the last three words “by talking flies” in the title since it is factually inaccurate and will be confusing to some readers.

[Editors’ note: a previous version of this study was rejected after peer review, but the authors submitted for reconsideration. The previous decision letter after peer review is shown below.]

Thank you for choosing to send your work entitled “Social learning allows *Drosophila* to anticipate a predator-threat by triggering ovarian apoptosis” for consideration at *eLife*. Your full submission has been evaluated by Eve Marder (Senior editor) and four peer reviewers, one of whom is Peggy Mason, a member of our Board of Reviewing Editors, and the decision was reached after discussions between the reviewers. Based on our discussions and the individual reviews, we regret to inform you that this version of the manuscript will not be considered further for publication in *eLife*. While all reviewers found the topic of the manuscript fascinating, several of them felt that further experiments were needed to clarify the model proposed. As you know, *eLife* does not allow us to ask for significant new experiments at the revise stage. However the interest of your work is such that the reviewers wanted to encourage a new submission with the addition of a new experiment as detailed below. Of course, you are free to take this manuscript elsewhere at this time if you chose to do so.

This manuscript describes a series of potentially exciting results that demonstrate social communication in flies. A female fly that is exposed to a wasp stops laying eggs and another female exposed to this wasp-exposed fly in turn stops laying eggs without ever having been exposed to a wasp herself. A strength of the paper is the elucidation of the mechanism of oviposition suppression in the teacher and student flies. However, there are weaknesses which center on the identification of vision as the critical modality used in the communication from teacher to student.

A more compelling test of vision's involvement in the teacher-student communication is needed. It is suggested that a set of experiments are performed in the dark or in long wavelength light. While *ninaB* mutants are known to have defective vision, it is not clear that impaired vision is their only defect.

It is also recommended that non-normalized data be shown. Directly comparing the number of eggs not only will give the readers a sense about this paradigm's consistency in egg-laying rate but also increase ones' confidence in the suppression effect. For example, suppression from 30 eggs to 10 eggs is more compelling than suppression from 3 eggs to 1 egg.

Genotype controls need to be shown. It is standard practice to include the results of genotype controls (e.g., *OK107/+* and *UAS-TeTx/+*), so that one can be sure that the observed effect is not due to additive effects of parental insertions.

In addition, there are several points in the manuscript that either require further experiments or modification of the conclusions. These include:

1) IR-mediated olfaction has not been ruled out. *Or83b* is now known as *Orco*. When *Orco* is knocked out the fly is not anosmic as ionotropic olfactory receptors or *Ir* receptors remain.

2) The involvement of the MBs is not convincingly demonstrated. *OK107* is not a clean MB driver.

3) The learning genes that were studied have pleiotropic effects. Without complementary evidence to accompany the genetic findings, these results are not compelling.

Additional comments that the authors may want to consider in the future:

1) Can the teacher fly still teach pupils 48 hours after exposure, or is teaching limited to 24-48 hours? If not, this may suggest that information has a half-life, and that its persistence is independent of the individual that initially observed it. If that's true, a pupil may be able to become a teacher if it is allowed to teach at an earlier time point.

2) While the removal of the wings clearly affects behavior, and this could be how information is being transferred to the student, this manipulation likely alters many aspects of behavior. Are the flies less mobile? Do they lay differently? It would be useful to parse this out.

3) What happens if the teachers are in the minority relative to the students? For example: use 3 teachers and 6 naive flies. Would their social influence be diluted out in a larger group? This may imply that a naive fly is “considering” the relative weights of social input to make a decision.

---

## [Author Response]

*1) In a number of places, the data presented in a panel are clearly the wrong data i.e. it is a copy of other panels found in other figures. I noted this in*
Figure 8—figure supplement 1
*and*
Figure 9*, but there may be other instances of this. The authors need to go through the figures very carefully to make sure they are all correct*.

Thank you very much for catching our mistake. The labels for teacher and student in these graphs were reversed. We have triple checked our figures and the data is correct, and the labels have been fixed accordingly in Figure 8—figure supplement 1 and Figure 9 in addition to Figure 3 (all which had reversed labels). We have thoroughly examined the other figures to ensure this mistake is not present elsewhere in the manuscript.

*2) There are some irregularities in the way genotypes are expressed. “>” usually implies a two-tiered system like GAL4 driving a transgene's expression i.e. two transgenes. In a number of the figures, this symbol is used in a non-canonical way to indicate a single transgene. An example is*
Figure 13—figure supplement 1
*where the labels say “*UAS>TeTx” *when I believe what is trying to be indicated is a* UAS-TeTx/+ *genotype. All the figures need to be edited to correct this*.

We have corrected the nomenclature used in our manuscript figures. For instances where “>” was used, we have replaced with “-” (i.e. *UAS-TeTx*) in both titles and labels of teachers and students (see Figure 13—figure supplement 1, Figure 14—figure supplement 1, and Figure 15—figure supplement 1). We have changed the text accordingly. Thank you for pointing this out to us.

*3) Is there information on how flies were reared, aged, stored prior to assays? If I couldn't find it, no problem. If it's not there, it should be*.

We previously had a line of text explaining rearing and aging prior to the experiment. We feel a more in depth description detailing our fly preparation and husbandry would be beneficial. See text changes in the subsection headed “Fly Oviposition” that now emphasizes this point and states:

“Flies aged 3-5 days post-eclosion on fresh, molasses-based, *Drosophila* media were used in all experiments. […] We stress the importance of aging the flies on fresh media, as it appears that flies aged on old media (i.e. the same media in which they eclosed) are nutrient deprived and naturally lay very few eggs.”

[Editors’ note: the author responses to the previous round of peer review follow.]

As suggested in your decision letter, we have carefully considered the reviewers' comments and would be very grateful if you would consider this new submission. This is a much revised story with additional/new experiments that address all the reviewers' concerns. Below, please see our responses to the reviewers' concerns and suggestions and specific reference to how the text and/or data have been changed to reflect this.

The title of this new submission is “Predator-induced changes in behavior and physiology are socially transmitted from exposed to naïve flies.” We believe that this manuscript presents the first example of social communication between female *Drosophila* that is completely based on visual cues with physiological consequences (apoptosis) in germline cells. This change in germline development provides a molecular explanation for the change in reproductive behavior, thus establishing a link between the visual system and female germline cells. We also show that persistence of this change in reproductive behavior requires a functional mushroom body and wild-type functions of learning and memory genes, again demonstrating a link between adult brain function and germline cell physiology. Because both behavioral and physiological changes are triggered by an ecologically relevant predator, we suggest that *Drosophila* evolved these specific innate behavioral and physiological responses to this serious environmental threat. Therefore, we suggest that this story would have broad interest to the *eLife* readership.

*This manuscript describes a series of potentially exciting results that demonstrate social communication in flies. A female fly that is exposed to a wasp stops laying eggs and another female exposed to this wasp-exposed fly in turn stops laying eggs without ever having been exposed to a wasp herself. A strength of the paper is the elucidation of the mechanism of oviposition suppression in the teacher and student flies. However, there are weaknesses which center on the identification of vision as the critical modality used in the communication from teacher to student*.

*A more compelling test of vision's involvement in the teacher-student communication is needed. It is suggested that a set of experiments are performed in the dark or in long wavelength light. While* ninaB *mutants are known to have defective vision, it is not clear that impaired vision is their only defect*.

Done. Please see Figure 8 and Figure 8—figure supplement 1 for experiments that took place in the dark or a dark-light combination. Please also see Figure 9 for experiments where only visual interaction was allowed between flies and wasps and teachers and students.

*It is also recommended that non-normalized data be shown. Directly comparing the number of eggs not only will give the readers a sense about this paradigm's consistency in egg-laying rate but also increase ones' confidence in the suppression effect. For example, suppression from 30 eggs to 10 eggs is more compelling than suppression from 3 eggs to 1 egg*.

Done. Please see text and Supplementary files 17 and 18 for corresponding raw numbers.

*Genotype controls need to be shown. It is standard practice to include the results of genotype controls (e.g.,* OK107/+ *and* UAS-TeTx/+*), so that one can be sure that the observed effect is not due to additive effects of parental insertions*.

Done. Please see Figure 13—figure supplement 1 and Figure 15—figure supplement 1.

*In addition, there are several points in the manuscript that either require further experiments or modification of the conclusions. These include*:

*1) IR-mediated olfaction has not been ruled out.* Or83b *is now known as* Orco*. When* Orco *is knocked out the fly is not anosmic as ionotropic olfactory receptors or* Ir *receptors remain*.

The text has been modified. Additional experiments have also been performed to address the role of visual cues. Please see Figure 9. Briefly, first we conducted wasp-fly and teacher-student interactions in complete darkness and this led to no transmission of information from wasp-to-fly or from teacher-to-student; second, we constructed glass chambers where flies could see wasps or teachers through the glass but could not smell, touch or taste either wasp or teachers. In contrast to the exposure in the dark, these experiments allowed transmission of information from wasp-to-flies and teachers-to-students.

*2) The involvement of the MBs is not convincingly demonstrated.* OK107 *is not a clean MB driver*.

We have performed the experiment involving blocking synaptic transmission with *UAS-TeTx* with a second MB driver (MB 247). Please see Figure 13—figure supplement 1. We have included additional experiments using the MB Gene-Switch line as well. Please see Figure 15 and Figure 16 and Figure 15—figure supplement 1 and Figure 16—figure supplement 1.

*3) The learning genes that were studied have pleiotropic effects. Without complementary evidence to accompany the genetic findings, these results are not compelling*.

We have tested additional alleles of the genetics mutants originally tested (*rut, dnc, Adf1, orb2*). We have also tested two additional mutants with two alleles of each (*amn* and FMR1). Please see Figure 12 and Figure 12—figure supplement 1. We narrowed the scope to examine the role of Orb2 and expressed an RNA-hairpin against Orb2 using the OK-107 MB driver and the MB Gene-Switch line to specifically knockdown *orb2*. These RNAi results recapitulate the mutant results. Using the Gene- Switch line allows us to activate the RNA-hairpin construct after development has finished, minimizing pleiotropic results. Please see Figure 14, Figure 15, Figure 16 and Figure 14—figure supplement 1, Figure 15—figure supplement 1 and Figure 16—figure supplement 1.

We suggest that it is highly unlikely that multiple alleles of six (6) different genes (*rut*, *dnc*, *Adf1*, *orb2* and FMR1) all have the same type of pleiotropic effects. Additionally, that such pleiotropic effects be phenocopied by driving RNAi-hairpin targeting Orb2 (but not controls such as the white gene) in a specific tissue would be unprecedented. Lastly, that all these mutants and the RNAi lines are perfectly capable of perceiving wasps, reducing oviposition and triggering apoptosis strongly suggests that these mutants are not general and non-specific disruptors of behavior. Instead, we suggest that a more likely interpretation is that the observations we report are indeed indicative of bona fide memory of a learned experience, and that disruption of these genes affect learning and memory functions of the brain. We urge the reviewers to consider the use of these genetic tools (mutants and RNAi) in the context of all the other assays reported herein, and how very unlikely it would be that these observations were due to pleiotropic effects.

*Additional comments that the authors may want to consider in the future*:

*1) Can the teacher fly still teach pupils 48 hours after exposure, or is teaching limited to 24-48 hours? If not, this may suggest that information has a half-life, and that its persistence is independent of the individual that initially observed it. If that's true, a pupil may be able to become a teacher if it is allowed to teach at an earlier time point*.

Done. Please see Figure 3 demonstrating teacher ability to teach multiple student cohorts.

*2) While the removal of the wings clearly affects behavior, and this could be how information is being transferred to the student, this manipulation likely alters many aspects of behavior. Are the flies less mobile? Do they lay differently? It would be useful to parse this out*.

We have examined this question using flies mutant in *ewg*, which yields flies with fully formed wings, though they are not able to move their wings. Please see Figure 11. We further confirmed that the *ewg* function required for the teaching behavior was wing specific by rescuing the neuronal function but not the muscle function of *ewg* (see Figure 11).

*3) What happens if the teachers are in the minority relative to the students? For example: use 3 teachers and 6 naive flies. Would their social influence be diluted out in a larger group? This may imply that a naive fly is “considering” the relative weights of social input to make a decision*.

We feel that this is a very interesting point, but at the time is outside of the scope of the paper. We wish to address this ratio and other ways to dilute the signal in future publications.